# Deep Tabular Learning via Distillation and Language Guidance

**Ruohan Wang**                                                    *wang_ruohan@i2r.a-star.edu.sg*
*Institute for Infocomm Research (I2R), A\*STAR, Singapore*

**Wenhao Fu**                                                        *fu_wenhao@i2r.a-star.edu.sg*
*Institute for Infocomm Research (I2R), A\*STAR, Singapore*

**Carlo Ciliberto**                                                      *c.ciliberto@ucl.ac.uk*
*AI Center, University College London*

**Reviewed on OpenReview:** *https://openreview.net/forum?id=p6KIteShzf*

## Abstract

Tabular data is arguably one of the most ubiquitous data structures in application domains such as science, healthcare, finance and manufacturing. Given the recent success of deep learning (DL), there has been a surge of new DL models for tabular learning. However, despite the efforts, tabular DL models still clearly trail behind tree-based approaches. In this work, we propose DisTab, a novel framework for tabular learning based on the transformer architecture. Our method leverages model distillation to mimic the favorable inductive biases of tree-based models, and incorporates language guidance for more expressive feature embeddings. Empirically, DisTab outperforms existing tabular DL models and is highly competitive against tree-based models across diverse datasets, effectively closing the gap with these methods.

## 1 Introduction

Deep learning (DL) has achieved remarkable progress in learning from visual (He et al., 2016; Dosovitskiy et al., 2020), textual (Vaswani et al., 2017; Brown et al., 2020) or audio data (Arik et al., 2018; Qin et al., 2023), emerging as the preferred approach for various tasks such as image classification Deng et al. (2009) and language translation Poibeau (2017). Inspired by these successes, there is a surge of interest in extending DL capabilities to *tabular learning*. Tabular data stands out as one of the most ubiquitous data structures for diverse domains, from patient records in healthcare to experimental results in scientific research. The ability to effectively glean insights from tabular data holds immense significance with many applications.

Among the existing works on tabular DL, a major research direction adopts model pre-training by initially bootstrapping DL models through *pre-text (or pre-training) tasks* before training (or fine-tuning) them on the actual labeled data of interest. Pre-text tasks tailored for tabular learning includes contrastive learning, input reconstruction and/or data synthesis via random perturbations of the labeled data (Ucar et al., 2021; Bahri et al., 2022; Majmundar et al., 2022). Orthogonally, Wang & Sun (2022); Ye et al. (2024); Yan et al. (2024) proposed cross-table pre-training approaches to learn features that can generalize across different tabular datasets. Empirical evidence supports the efficacy of pre-training, demonstrating superior generalization performance compared to direct training on the labeled data from scratch.

On the other hand, following the successes of transformer architectures (Vaswani et al., 2017) in challenging domains including natural language processing (NLP) (Ouyang et al., 2022), computer vision (CV) (Dosovitskiy et al., 2020), and reinforcement learning (RL) (Chen et al., 2021), recent works have investigated their impact on tabular applications (Gorishniy et al., 2021; Somepalli et al., 2021; Wang & Sun, 2022). The core idea behind these "tabular transformers" is to represent table columns (or features) as a sequence

of tokens, aligning with the input format required by transformers. Gorishniy et al. (2021) demonstrated that transformer architectures outperform classical models such as multi-layer perceptrons (MLP) on tabular learning benchmarks. Recent work also explored different strategies for embedding tabular features into token sequences, aimed at further improving model performance (Gorishniy et al., 2022; Wang & Sun, 2022).

While the recent interest in tabular DL has unarguably led to significant advancements on the topic, benchmark evaluations (Chen et al., 2023; Zhu et al., 2023; Shwartz-Ziv & Armon, 2022; Borisov et al., 2022) indicate that more traditional algorithms, such as gradient boosting decision trees (GBDT) (see for example Ke et al., 2017), remain the state-of-the-art for tabular learning. In particular, tree-based approaches demonstrate overall better generalization performance across diverse dataset sizes and exhibit robustness without requiring extensive hyper-parameter tuning, in clear contrast to tabular DL methods (Prokhorenkova et al., 2018). Consequently, Grinsztajn et al. (2022); Shwartz-Ziv & Armon (2022) have specifically investigated why DL trials behind tree-based methods, and bridging the performance gap between DL methods and tree-based ones remain an open and important challenge for tabular learning.

Despite these challenges, there exist compelling motivations for using DL models for tabular learning. Specifically, DL models could generate expressive tabular representations for downstream tasks (Grinsztajn et al., 2022) and are capable of knowledge transfer across different tabular datasets (Wang & Sun, 2022; Hollmann et al., 2022; Yan et al., 2024; Ye et al., 2024). For instance, Hollmann et al. (2022) allows the trained tabular model to solve small classification tasks efficiently with significant speed-up via knowledge transfer. Moreover, DL allows for more efficient integration of different input modalities or information sources to enhance model capabilities (as demonstrated in vision-language models (Alayrac et al., 2022; Zhang et al., 2021)).

To address these limitations, we introduce DisTab, a new tabular DL framework aimed at bridging the gap with tree-based approaches. DisTab leverages knowledge distillation (Hinton et al., 2015) for pre-training, directly employing a suitable tree-based model as the teacher. This enables our model to emulate the favorable inductive biases inherent in tree-based approaches (Grinsztajn et al., 2022), effectively closing the performance disparity between tabular DL and GBDTs.

Furthermore, DisTab showcases how tabular DL models can capitalize on the capabilities of neural architectures (in particular transformer models) to organically incorporate different information sources without need for ad-hoc model design. In particular, we introduce the concept of *language-guidance* for DisTab to integrate available textual information (e.g., textual descriptions for column headers or categorical features) for embedding tabular features. We show that this choice enhances the conventional tabular embeddings with semantic context to improve generalization performance. This integration serves two purposes: from the practical perspective, it enables DisTab to achieve state-of-the-art performance on tabular data. From the methodological perspective, it showcases the benefits of tabular transformers as a natural architecture to integrate different (meta) information available for tabular learning. In contrast, tree-based methods are relatively more rigid and the integration of unstructured information such as language would be less straightforward.

Empirically, we conduct a extensive comparison of DisTab against existing tabular learning approaches across diverse tabular datasets. Our results demonstrate that DisTab not only outperforms existing tabular DL methods but also achieves competitive performance against GBDT models. Furthermore, we conduct comprehensive ablation studies on DisTab, where we systematically analyze the contributions of each of its components. Our findings consistently indicate that knowledge distillation and language guidance both play crucial roles in enhancing model performance.

The contributions of this paper are summarized as follows: **1)** we introduce DisTab, a novel framework for tabular deep learning, incorporating knowledge distillation for pre-training and language guidance for feature embedding. **2)** Our framework outperforms existing tabular DL methods and stands as a competitive alternative to GBDT models. **3)** We present extensive ablation experiments to study the impact of individual model components, offering valuable insights for exploring new avenues in building tabular DL models.

## 2   Related Works

**Tabular Pre-training.** Inspired by the success of pre-training in CV and NLP, recent studies have explored applying these strategies to tabular data settings, where training data scarcity is a significant concern. (e.g., Bahri et al., 2022; Yoon et al., 2020; Majmundar et al., 2022; Rubachev et al., 2022; Zhu et al., 2023). Among these approaches, Ucar et al. (2021) introduced an auto-encoder model equipped with an objective function to reconstruct randomly masked columns of a table. Bahri et al. (2022) adapted contrastive learning as the pre-training objective for tabular tasks, extending the SimCLR framework (Chen et al., 2020) originally designed for visual representation learning. Furthermore, Rubachev et al. (2022); Wang & Sun (2022) integrated "target-aware" pre-training objectives by incorporating target labels, resulting in performance enhancements.

The majority of existing pre-training approaches are domain-specific: the labeled training data also serve as pre-training data (Bahri et al., 2022; Ucar et al., 2021), or they are closely related Wang & Sun (2022). In contrast, Zhu et al. (2023) showcased the viability of pre-training on a large collection of tables spanning diverse domains through multi-task learning Sener & Koltun (2018). In this paradigm, each tabular learning task possesses its independent feature embeddings and objective functions, while sharing a tabular transformer model trained to generalize across different tabular datasets.

Our proposed DisTab also adopts domain-specific pre-training but opts for knowledge distillation (Hinton et al., 2015) as the pre-training objective. Like existing pre-training methodologies, DisTab leverages synthetic training samples generated during pre-training to mitigate data scarcity. However, it possesses the added potential to learn the inductive biases of tree-based models favorable for tabular tasks, as hypothesized in Grinsztajn et al. (2022).

**Tabular transformers.** Transformer models (Vaswani et al., 2017) have recently gained significant popularity in tabular learning scenarios. For instance, Gorishniy et al. (2021) introduced FT-Transformers, demonstrating superior performance in tabular classification/regression tasks compared to classical DL architectures like MLPs and ResNets. Additionally, Somepalli et al. (2021) proposed column-wise attention to capture inter-sample interactions, while Fastformer utilizes additive attention on tabular tasks, offering a lightweight attention mechanism with linear complexity relative to the length of input sequences (Wu et al., 2021).

A crucial aspect in designing tabular transformers is how to embed tabular features into token sequences, aligning with the input format required by transformer models. Most existing approaches (Zhu et al., 2023; Somepalli et al., 2021; Gorishniy et al., 2021) use a single token to represent each column and learn linear mappings from raw tabular features to token embeddings. Conversely, Wang & Sun (2022) map each column as a variable number of tokens, with column headers and text for categorical features represented by multiple tokens based on their word counts. Moreover, (Gorishniy et al., 2022) investigated piece-wise linear encoding and periodic encoding for numerical features, demonstrating improved generalization over linear mappings. Our DisTab adopts a transformer architecture, however it differs from the existing approaches by introducing language-guided embeddings, which encode textual information for columns as context tokens to augment previously proposed feature embeddings (e.g., learned linear mappings).

**GBDTs for Tabular Learning.** Despite the advancements in tabular DL methods, recent large-scale benchmarks have demonstrated that gradient boosting decision tree (GBDT) models remain the state-of-the-art for tabular learning (Grinsztajn et al., 2022; Chen et al., 2023; Zhu et al., 2023). Commonly used GBDT models include XGBoost (Chen & Guestrin, 2016), CatBoost (Prokhorenkova et al., 2018), and LightGBM (Ke et al., 2017). They also offer several advantages, such as interpretability, the capability to handle heterogeneous features including null values, and robustness without hyper parameter tuning. However, a critical challenge of these models is to properly integrate large-scale vectorial data (such as text or image representation from a vision or language model respectively). For instance, while Ye et al. (2024); Wang & Sun (2022); Yan et al. (2024) have demonstrated the effectiveness of incorporating textual

headers into tabular deep learning (DL) models to improve generalization performance, it is neither trivial nor clear how to integrate such information in tree-based models. This creates an opportunity for tabular DL approaches to compete in scenarios where the input data includes modalities typically represented as vectors (e.g., column header information in tabular datasets). Given the reliable performance of GBDTs on tabular data, DisTab directly utilizes trained tree-based models for pre-training, employing them as teacher models. We evaluate DisTab alongside tree-based models to compare their relative performance.

## 3   Method

In this section, we detail the key components for DisTab, including the proposed feature embeddings with language guidance in Sec. 3.1, the pre-training process in Sec. 3.2 and the overall algorithm in Sec. 3.3.

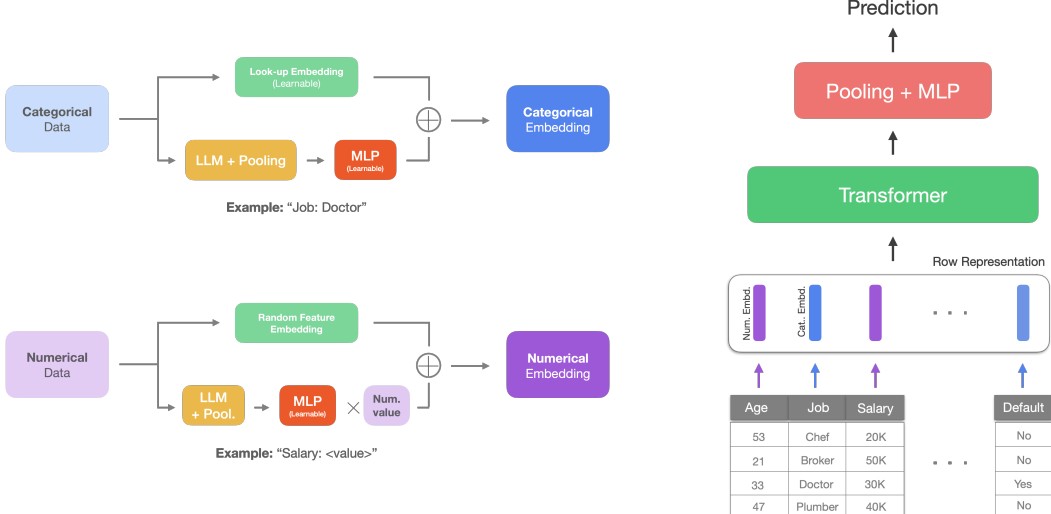

Figure 1: **Left:** Language-guided categorical (**Top**) and numerical (**Bottom**) embeddings for DisTab. **Right:** DisTab architecture. A categorical or numerical embedding is extracted from individual column entry, processed by a transformer and an MLP before returning a prediction.

**Notation.** For a given supervised learning problem on tabular data, we denote the dataset as $\mathcal{D} = \{x^i, y^i\}_{i=1}^N$ where $x^i \in \mathbb{X}$ is a row in the table and $y^i \in \mathbb{Y}$ the corresponding label. Let $X_j$ represent the $j$-th column of $\mathcal{D}$, $h_j$ the text header (if available) for the $j$-th column, and $x_j^i$ the $j$-th column of $x_i$. Lastly, we denote $f_{\text{enc}} : \mathcal{T} \to \mathbb{R}^d$ as the text embedding function (e.g., BehnamGhader et al., 2024) that maps a text string $t \in \mathcal{T}$ to a continuous embedding $f_{\text{enc}}(t) \in \mathbb{R}^d$.

### 3.1   Input Embeddings for Tabular Features

Similar to previous works (Gorishniy et al., 2021; 2022; Zhu et al., 2023; Somepalli et al., 2021). we choose to represent each column with a single token. We observe that this design, when coupled with a suitable transformer architecture (Sec. 3.3), satisfies output invariance for all permutations of a input sequence. We argue that this is a potentially desirable inductive bias for tabular data, since permutations of a table's columns should not affect the underlying learning task.

**Language-Guided Embeddings.** Most existing approaches for tabular learning overlook valuable textual information embedded within tabular datasets, such as column headers or text descriptions associated with categorical values. These textual elements provide rich semantic context, which could lead to better input representation and in turn improved performance. Tabular data is a human construct and textual column headers are often necessary even for human interpretation. It is thus reasonable to assume the presence of

meaningful headers and to allow ML models access to them. Therefore, we propose to augment input tokens for each table column with available textual information.

**Categorical Embedding.** We embed a categorical feature $x_j^i$ as follows,

$$E_j(x_j^i) = m_j(x_j^i) + f_{\text{enc}}(h_j \oplus x_j^i) \tag{1}$$

where $m_j : \mathcal{T} \to \mathbb{R}^d$ denotes the learnable embedding function that embeds $x_j^i$ via look-up, commonly adopted in tabular transformer models (Zhu et al., 2023; Wu et al., 2024; Gorishniy et al., 2021). The term $f_{\text{enc}}(h_j \oplus x_j^i)$ provides semantic context for the feature by concatenating the column header $h_j$ and the textual description for $x_j$ as a text string to be embedded by $f_{\text{enc}}$. We combine the two embeddings additively to derive the final embedding for $x_j^i$. The embedding process is depicted in Fig. 1: the lookup embeddings (top stream, denoted in green) is linearly combined with the language-based embedding $f_{\text{enc}}$ (bottom stream). We define $f_{\text{enc}} = f_{\text{proj}} \circ f_{\text{lm}}$, where $f_{\text{lm}} : \mathbb{R}^{n \times d} \to \mathbb{R}^{n \times d}$ outputs the sequence representation $f_{\text{lm}}(t) = v \in \mathbb{R}^{n \times d}$ from a language model. Then, $f_{\text{proj}} = \text{MLP}(\text{AvgPool}(v))$ summarizes the sequence representation into a single vector, followed by a small learnable projection network. Please see App. B.2 for further model details.

**Numerical Embedding.** We utilize a periodic activation function $p_\sigma(\cdot)$ for encoding numerical features. Specifically, we define $p_\sigma(\cdot)$ as follows:

$$p_\sigma(x) = \sin(v) \oplus \cos(v) \in \mathbb{R}^k \qquad \text{where} \qquad v = [c_1 x, \ldots, c_{k/2} x], \quad \text{with} \quad c_i \sim \mathcal{N}(0, \sigma^2) \tag{2}$$

where $\sin(v)$ and $\cos(v)$ apply the corresponding function entry-wise to $v$. Notably, Rahimi & Recht (2008; 2007) showed that $p_\sigma(x)$ is a feature map that approximates the Gaussian RBF kernel, namely $p_\sigma(x) \cdot p_\sigma(y) \approx \exp(-\frac{\sigma^2(x-y)^2}{2})$ (with the approximation improving as the latent dimension $k$ in (2) increases). $p_\sigma(x)$ is also known as *random features* in the kernel literature Rahimi & Recht (2007). Evidently, the bandwidth $\sigma$ is crucial to the quality of encoding, as it determines how "similar" two values $x, y$ should be regarded within a table column.

Using $p_\sigma(\cdot)$, we embed a numerical feature $x_j^i$ as follows,

$$E_j(x_j^i) = \bigoplus_{\sigma \in \Sigma} p_\sigma(x_j^i) + f_{\text{enc}}(h_j) \times x_j^i \tag{3}$$

where $\Sigma = \{\sigma_1, \ldots, \sigma_m\}$ is a set of bandwidths. To account for different length scales for similarity, we use $\bigoplus_{\sigma \in \Sigma} p_\sigma(x_j^i)$ to concatenate multiple $p_\sigma(x_j^i)$ with different $\sigma$ into a single embedding of dimension $d$ with $k = \frac{d}{|\Sigma|}$ in (2). Our formulation differs from Gorishniy et al. (2022) that uses only a single $p_\sigma(\cdot)$ for encoding, but requires expensive hyper-parameter tuning for $\sigma$. In our experiments, we set $\Sigma = \{0.1, 1, 10\}$.

The embedding process modeled in (3) is depicted in Fig. 1: the random-feature embedding (top stream, depicted in red) is linearly combined with the language model embedding $f_{\text{enc}}$ of the header (bottom stream), scaled proportionally to the numerical feature that is being embedded. This latter quantity provides semantic context to encode the associated numerical value suitably.

**Relation to Previous Works.** Our embedding functions generalize previous works by incorporating language guidance. Specifically, we recover the embedding functions in Gorishniy et al. (2021; 2022); Zhu et al. (2023); Somepalli et al. (2021) if we remove $f_{\text{enc}}(\cdot)$ from (1) and (3). Another key difference is that our embedding functions combine different representations of a given feature additively to derive a unified embedding, which we find to work well in practice.

### 3.2 Model Pre-training

It is well established that pre-training improves the generalization performance of tabular DL methods (Ucar et al., 2021; Bahri et al., 2022; Zhu et al., 2023). We argue that there are two key factors contributing to the

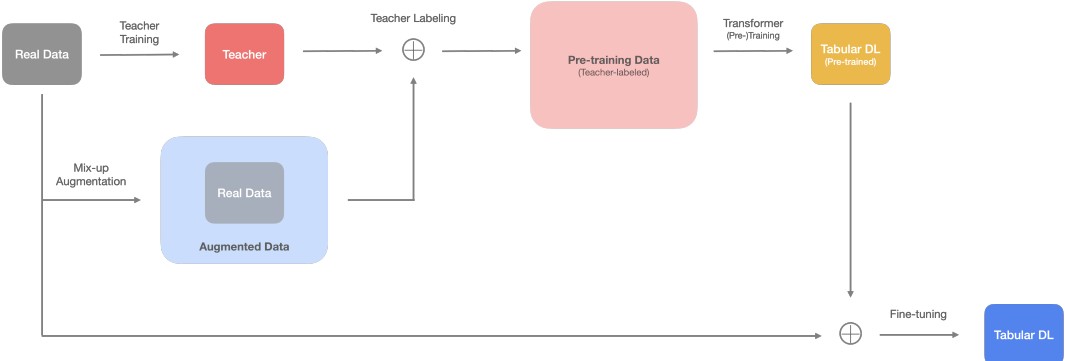

Figure 2: The distillation-based pre-training pipeline for DisTab.

improved performance. Most directly, pre-training synthesizes new training data from real one, significantly increasing the size of the training data and in turn improves generalization performance. This is particularly crucial for small tabular datasets, for which DL models tend to overfit.

On the other hand, pre-training provides auxiliary learning signals to improve generalization performance. As discussed in Sec. 2, commonly used self-supervised objectives include reconstruction loss and contrastive loss using random perturbation of real data. For reconstruction learning, we observe that some table columns cannot be learned from others. For instance, tables involving transaction data (e.g., house sales) often include sales date as a column, which cannot be reliably predicted from other columns describing product features (e.g., house size or room count). This may limit the effectiveness of reconstruction learning. For contrastive learning, heuristics are required to generate similar views of the same data, which may not align with the desired data similarity with respect to the actual learning task. For instance, random perturbation of a critical column (e.g., lab results in medical diagnosis) could drastically change the target label, but is viewed as a similar view since only a single column is perturbed.

**Distillation for Pre-training.** We therefore propose to utilize knowledge distillation of tree-based models for pre-training. Knowledge distillation aims to directly mimic the superior performance of tree-based models in tabular settings, without needing to heuristically constructing a pre-text task with potential limitations discussed above. Formally, given a tabular dataset $\mathcal{D} = (x^i, y^i)_{i=1}^N$ and access to a teacher model $g_T$, we denote the teacher-labeled dataset as $g_T(\mathcal{D}) \triangleq (x^i, g_T(x^i))_{i=1}^N$. We also modifies the mix-up technique (Zhang et al., 2017) for data augmentation $\text{Aug}(\mathcal{D})$,

$$x \sim \text{Aug}(\mathcal{D}) \text{ where } x = x^i \odot m + x^j \odot (1 - m), m_k \sim B(1, 0.5) \tag{4}$$

where $\odot$ denotes point-wise multiplication, and $B(1, 0.5)$ a Bernoulli distribution with probability 0.5. The augmentation synthesizes new samples by randomly mixing columns, as determined by $m$, from 2 random samples $x^i, x^j$ from the real data.

With the notations introduced above, the distillation pre-text task is

$$\arg\min_{\theta} \sum_{(x,y) \in g_T(D \cup \text{Aug}(D))} \ell(f_\theta(x), y) \tag{5}$$

where $\ell$ is the suitable loss function for the tabular task, such as least-square errors for the regression setting.

By combining model distillation and mix-up augmentation, we are able to sidestep heuristics needed to define target labels for synthetic samples. For instance, the original mix-up approach determines labels for synthetic samples using $y^i, y^j$ and $m$. Instead, our approach labels all real and synthetic samples with the teacher model and pre-train on this dataset to emulate the teacher's behavior.

The pre-training pipeline described above is summarized in Fig. 2: first, the teacher is trained on the ground-truth data. In all experiments we choose CatBoost as the teacher model, which has been observed to

achieve consistently state-of-the-art performance (in particular average ranking) according to recent benchmarks (Zhu et al., 2023; Chen et al., 2023) and our experiments in Sec. 4. The ground-truth (input) data is then augmented via mix-up in (4) and labeled by the teacher. The augmented dataset is used to pre-train a tabular DL architecture, followed by fine-tuning solely on the original data to yield the final model.

### 3.3 DisTab Algorithm

**Model Architecture.** Our model $f_\theta = f_{\text{out}} \circ f_{\text{trans}}$ consists of a transformer model $f_{\text{trans}} : \mathbb{R}^{n \times d} \to \mathbb{R}^{n \times d}$ that processes table rows and $f_{\text{out}} : \mathbb{R}^{n \times d} \to \mathbb{Y}$ that maps the transformer output to target labels. We use a simplified Llama transformer architecture (Touvron et al., 2023) for $f_{\text{trans}}$, which removes all position encoding. $f_{\text{out}}(x) = \text{MLP}(\text{AvgPool}(x))$ apply average pooling to the transformer output to obtain sample representations, followed by a linear layer for final output. We note that $f_\theta$ is invariant to all permutations of a given input, a desirable inductive bias for tabular data as discussed earlier.

For model training, we first transform the training data using the embedding functions (Sec. 3.1). followed by distillation pre-training (Sec. 3.2). Since pre-training only uses teacher labels, we further fine-tune the pre-trained model using on only the real data with original labels.

## 4 Experiments

In this section, we evaluate our proposed method against different tabular learning methods on test performance, over a diverse set of benchmark datasets. We first describe the experimental settings below, followed by the detailed results and discussion[1].

**Datasets.** We use 25 datasets from OpenML for all evaluations (see Appendix A for details). We follow the datasets used in Zhu et al. (2023), but focus on those with meaningful textual column headers, since they allow us to apply and evaluate the proposed language-guided embeddings. For each OpenML dataset, we use the default train/test splits defined by the OpenML library to ensure better reproducibility (10% data is reserved for testing for each split). For each training split, we randomly partition 90% of data for training and the rest for validation. All methods are trained and evaluated using the same splits.

**Data Preprocessing.** For DL approaches designed with MLP architectures, we follow the previous works (Bahri et al., 2022; Yoon et al., 2020) to represent categorical features by one-hot encoding. For transformer-based architectures, categorical features are represented using a ordinal encoding. For all DL-based approaches, numerical features are scaled by z-score. For tree-based approaches, we adopt the default prepossessing associated with each method in AutoGluon (Erickson et al., 2020), a tabular learning library that provides strong performance for tree-based methods.

**Model Training.** For DisTab, we use a batch size of 1024 for pre-training and 128 during fine-tuning. For the existing DL tabular methods, we use the batch size 128 for both pre-training and fine-tuning, as recommended in Bahri et al. (2022); Zhu et al. (2023). All DL-based methods use Adam optimizer with a learning rate of 1e-4, with a weight decay of 1e-5, following Gorishniy et al. (2021); Rubachev et al. (2022). Number of pre-training and fine-tuning epochs are empirically determined for each method, but remain consistent across different tasks. For DisTab, we use 30 epochs for pre-training and 20 for fine-tuning.

**Evaluation methods.** We divide the datasets into regression, binary classification and multi-class classification tasks. For model performance, we use root mean least square (RMS) for regression tasks, area under the receiver operating characteristic curve (AUC) for binary classification, and accuracy for multi-class classification. For each task, every model is trained and evaluated using the same 5 splits, and we use the average performance over the 5 splits as a model's task performance.

---

[1]Our code is available at `https://github.com/RuohanW/DisTab`

Given the diversity of tabular datasets evaluated, we use the following metrics to effectively compare different methods over all datasets:

*Win matrix.* Following Bahri et al. (2022), we report our findings in the form of a $M \times M$ matrix $W$ where the $(i, j)$-th entry $W_{i,j}$ denotes the ratio of datasets for which method $i$ outperformed method $j$. We present this information in fractional form to include the total number of tasks evaluated and as a heat map to highlight scale.

*Average ranks.* While win matrix effectively conveys pair-wise comparison, it does not directly provide an overall ranking of all methods. We follow Zhu et al. (2023) to report the more traditional average ranking of all methods across each task category.

*Task performance.* We also include raw task performances for each method that underlies the win matrix and average ranking. For brevity, these results are deferred to the Appendix C.

**Baselines.** We consider a wide range of existing tabular methods for comparison, including CatBoost (CAT) (Prokhorenkova et al., 2018), random forests (RF) (Breiman, 2001), LightGBM (GBM) (Ke et al., 2017), and XGBoost (XGB) (Chen & Guestrin, 2016) for tree-based methods. For DL approaches, we include FastAI tabular (FastAI) (Howard & Gugger, 2020), FT-Transformer (FTT) (Gorishniy et al., 2021), XTab (Zhu et al., 2023), Saint (Somepalli et al., 2021), VIME (Yoon et al., 2020), SCRAF (Bahri et al., 2022), SwitchTab (Switch) (Wu et al., 2024) and TP-BERTa (TBERT) (Yan et al., 2024). The DL approaches include both transformer and MLP architectures, along with different pre-training strategies and models learned from scratch.

## 4.1 Comparison with DL Methods

We compare DisTab against a diverse set of existing DL tabular methods in Tab. 1 and Fig. 3. The evaluated methods include both MLP and transformer architectures, various pre-training strategies, as well as baseline models trained from scratch (FastAI and FTT).

For average rankings in Tab. 1, DisTab clearly outperforms the existing methods, with over 0.5 rank higher than the 2nd best performing method. Similarly in Fig. 3, our proposed method outperforms the DL baselines in all settings, winning in over 75% tasks in all pairwise comparison.

| Task Type | FASTAI | MLP Vime | Scraf | FTT | XTab | Transformer Saint | SwitchTab | TBERT | DisTab |
|---|---|---|---|---|---|---|---|---|---|
| Regression | $4.8 \pm 2.8$ | $6.4 \pm 1.7$ | $6.0 \pm 2.5$ | $3.4 \pm 1.7$ | $3.9 \pm 2.1$ | $5.1 \pm 1.8$ | $5.5 \pm 1.4$ | $7.4 \pm 2.4$ | $\mathbf{2.6} \pm 2.4$ |
| Binary | $6.0 \pm 2.4$ | $4.9 \pm 2.5$ | $6.2 \pm 2.4$ | $4.1 \pm 1.9$ | $3.4 \pm 1.4$ | $4.4 \pm 2.0$ | $6.3 \pm 2.1$ | $6.7 \pm 2.9$ | $\mathbf{3.0} \pm 2.4$ |
| Multiclass | $8.1 \pm 0.6$ | $5.8 \pm 1.9$ | $4.5 \pm 0.8$ | $2.9 \pm 2.0$ | $5.1 \pm 2.3$ | $5.7 \pm 2.6$ | $6.2 \pm 1.6$ | $4.6 \pm 2.4$ | $\mathbf{2.0} \pm 2.1$ |
| Overall | $6.2 \pm 2.6$ | $5.6 \pm 2.2$ | $5.6 \pm 2.2$ | $3.5 \pm 1.9$ | $4.0 \pm 2.1$ | $5.0 \pm 2.2$ | $6.0 \pm 1.8$ | $6.3 \pm 2.8$ | $\mathbf{2.6} \pm 2.3$ |

Table 1: Comparison of tabular prediction performance between DisTab and other tabular DL methods. Average rank and its standard deviation reported for each method. DisTab outperforms the existing methods for all task categories and is overall best performing.

While existing pre-training strategies perform generally well for classification settings: they beat the trained-from-scratch baseline in majority of tasks in Fig. 3. Their performance on regression tasks are mixed. this result indicates that general pre-training strategies adapted primarily from visual learning tasks may not be suitable for tabular learning, as we argued in Sec. 3.2. In contrast, distillation pre-training shows its efficacy by consistently outperforming the trained-from-scratch baseline and other pre-training strategies.

Lastly, we note that transformer-based models noticeable outperform MLP-based ones, validating similar observations in Grinsztajn et al. (2022); Gorishniy et al. (2021). The results empirically supports our hypothesis that transformers provide good prior for tabular learning with its permutation invariance with respect to table columns.

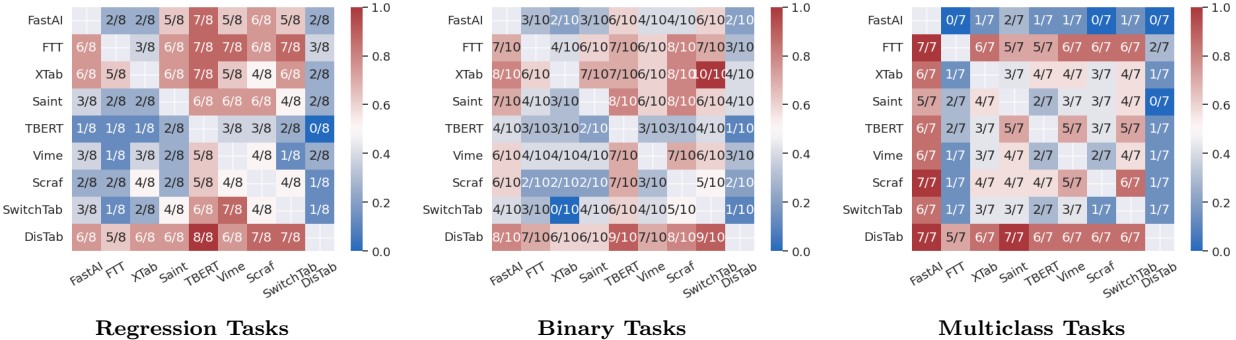

Figure 3: Win matrices between DisTab and existing DL tabular methods. DisTab outperforms all other methods in pairwise comparison.

## 4.2 Comparison with Tree-based Methods

To further assess DisTab's performance, we compared it against tree-based methods in Tab. 2. In terms of average rank, DisTab outperforms all tree-based methods for regression and multiclass tasks, while only trailing behind CatBoost for binary tasks. Our method is also the overall best performing over all tasks.

| Task Type | Metric | CAT | RF | GBM | XGB | DisTab |
|---|---|---|---|---|---|---|
| Regression | RMSE | $1.88 \pm 1.05$ | $4.12 \pm 1.05$ | $3.38 \pm 1.11$ | $3.88 \pm 0.78$ | $\mathbf{1.75} \pm 0.97$ |
| Binary | AUC | $\mathbf{1.90} \pm 0.54$ | $4.20 \pm 1.60$ | $3.80 \pm 0.60$ | $3.10 \pm 1.30$ | $2.00 \pm 0.89$ |
| Multiclass | Accuracy | $3.86 \pm 0.83$ | $3.71 \pm 1.48$ | $2.71 \pm 1.39$ | $2.57 \pm 1.05$ | $\mathbf{2.14} \pm 1.36$ |
| Overall | | $2.44 \pm 1.20$ | $4.04 \pm 1.43$ | $3.36 \pm 1.13$ | $3.20 \pm 1.20$ | $\mathbf{1.96} \pm 1.08$ |

Table 2: Comparison of tabular prediction performance between DisTab and tree-based methods. Average rank and its standard deviation reported for each method.

Consistent with previous benchmark results from Zhu et al. (2023); Chen et al. (2023), we re-validated CatBoost as the overall best performing tree-based method, which we used as the teacher model. For both regression and multiclass settings, DisTab outperforms the teacher model, suggesting that it is not merely "parroting" predictions from the teacher. In particular, CatBoost is in fact the worst performing model for multiclass setting. Despite distilling from a clearly sub-optimal model during pre-training, DisTab eventually emerged as the best performing model is this setting, indicating the robustness of the proposed approach.

Fig. 4 expands Tab. 2 to focus on pairwise comparison via the win matrices. For regression and binary setting, DisTab performs comparably to CatBoost and dominates the other tree-based approaches. For multi-class setting, DisTab clearly outperform random forests and CatBoost and is marginally better than LightGBM and XGBoost.

We highlight that the evaluated datasets not only include different task settings, but with diverse dataset sizes (from only 1k to over 580k samples), column counts (from 7 to 80) and presence of missing values. Across these datasets, Tab. 2 and Fig. 4 indicate that DisTab either surpasses or performs comparably to tree-based methods, effectively bridging the performance gap between tabular DL and tree-based approaches.

In Appendix C, we compare DisTab with tree-based methods using average task performance, to further quantify the scale of performance difference among them. The results are consistent with those in Tab. 2, showing that DisTab matches or surpass tree-based methods on the evaluated datasets.

## 4.3 Ablation Study

In this section, we investigate the efficacy of key components in DisTab, including distillation pre-training, language-guided embedding, and supervised fine-tuning. There are in total 6 valid combinations of the

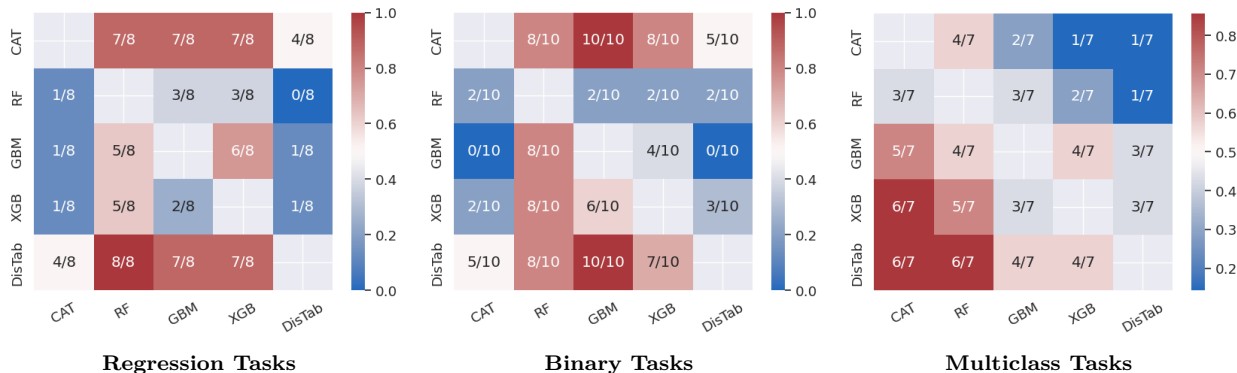

Figure 4: Win matrices between DisTab and tree-based methods.

above components and we compare their test performance in Tab. 3 and Fig. 5. As described in Sec. 3.1, model variants without "+LM" use standard input embeddings for tabular datasets, including learnable lookup embeddings and random features. The remaining models **additionally** include language-guided embeddings to provide additional context to input data. We also include CatBoost, the teacher model, in Fig. 5 as a comparison reference.

| Model | Distillation | LangEmbed | Finetune | Regression | Binary | Multiclass | All |
|---|---|---|---|---|---|---|---|
| Base | | | ✓ | $4.38 \pm 2.00$ | $3.90 \pm 1.14$ | $4.29 \pm 1.48$ | $4.16 \pm 1.57$ |
| Base+LM | | ✓ | ✓ | $4.62 \pm 1.58$ | $4.60 \pm 1.28$ | $4.21 \pm 1.89$ | $4.50 \pm 1.57$ |
| Distil | ✓ | | | $3.62 \pm 1.49$ | $3.95 \pm 1.56$ | $4.57 \pm 1.29$ | $4.02 \pm 1.51$ |
| Distil+LM | ✓ | ✓ | | $3.38 \pm 1.49$ | $3.40 \pm 1.69$ | $4.14 \pm 0.83$ | $3.60 \pm 1.47$ |
| FT | ✓ | | ✓ | $\mathbf{2.50} \pm 1.32$ | $2.65 \pm 1.52$ | $2.29 \pm 0.88$ | $2.50 \pm 1.31$ |
| DisTab | ✓ | ✓ | ✓ | $\mathbf{2.50} \pm 0.87$ | $\mathbf{2.50} \pm 1.91$ | $\mathbf{1.50} \pm 0.60$ | $\mathbf{2.22} \pm 1.41$ |

Table 3: Ablation comparison on DisTab. Each variant has one or more components turned off with respect to DisTab. Base model denotes standard supervised learning on the datasets. DisTab outperforms all variants, suggesting that pre-training, language embedding and fine-tuning all contributed meaningfully to model performance.

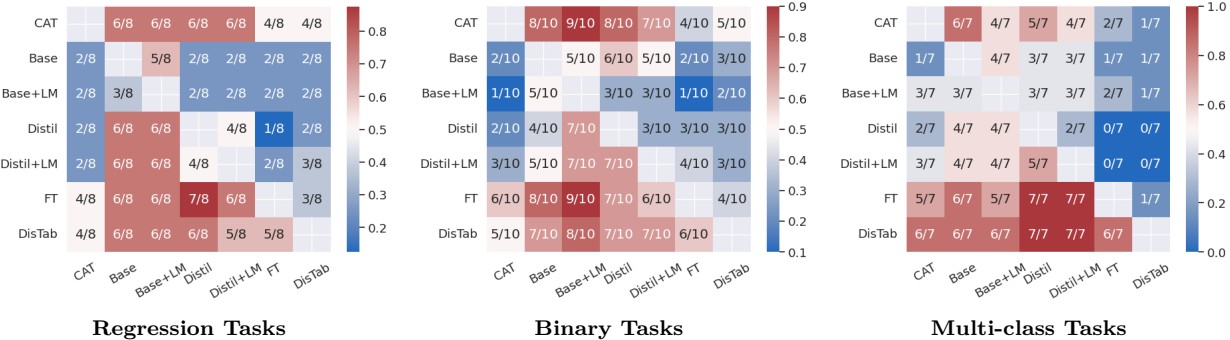

Figure 5: Win matrices between DisTab and its variants, with one or more components disabled. DisTab outperforms all DL variants, and achieves parity or surpasses the CatBoost teacher model.

**Distillation.** Tab. 3 indicates that distillation improves model performance: Distil beats Base in 14 out of the 25 tasks, while Distil+LM beats Base+LM in 17 tasks. However, the impact of distillation varies across different task settings. it is most beneficial for regression to improve 6 out of 8 tasks from Base,

but provides mixed results for classification. The results suggest that distillation of tree-based methods alone may be insufficient to reliably improve tabular DL models. However, FT, which combines distillation and fine-tuning, clearly outperforms Base, winning 20 out of the 25 tasks. The improvement also becomes consistent across all task settings. This strongly suggests the efficacy of distillation as a pre-training strategy to offer a robust model prior, and the necessity of fine-tuning for improved performance.

**Language-guided Embedding.** In Tab. 3, DisTab is able to improve the average rank over all tasks from 2.5 (obtained by FT) to 2.2 by incorporating semantic information in model input. The table also shows that language guidance performs robustly across all task settings and is particularly effective for multi-class setting, with an increase of 0.79 in average rank. Fig. 5 shows similar results when focused on pairwise comparison between DisTab and FT: the former outperforms the latter in 18 out of 25 tasks. The results clearly indicate the effectiveness of language guidance for tabular prediction.

Without pre-training, we observe that Base+LM underperforms Base. This effect is due to degraded performance on tasks of smaller size, since the added degrees of freedom introduced by language-guidance impact negatively Base+LM model, which is already prone to overfitting on small data volumes (see Sec. 3.2). The results suggest that language guidance is most useful when given sufficient training data, either leveraging synthetic data from pre-training, or using a training set of adequate size.

## 5    Discussion and Conclusion

In this work, we introduced DisTab as a framework to bridge the performance gap between DL and tree-based methods for tabular prediction tasks. We demonstrated that a straightforward yet previously overlooked strategy is to leverage distillation for model pre-training, employing an appropriate tree-based model as teacher. We also introduced a simple yet effective data augmentation strategy compatible with distillation to tackle training data scarcity, a common scenario in tabular domains. Empirically, our results suggest that DisTab compares favorably to other DL methods, including a variety of alternative pre-training strategies, model architectures customized for tabular learning, and learning from scratch. More importantly, our approach either surpasses or matches the performance of tree-based methods, effectively closing the performance gap between the two classes of methods.

Beyond these practical contributions, our work demonstrates also the potential structural advantages of tabular DL over tree-based methods, namely the flexibility and ease of integrating different information sources during learning, such as language. In particular, we showed how to incorporate semantic information during learning, including column headers and textual descriptions for categorical data. Our results show that language guidance is effective for tabular learning, improving the model performance for the vast majority of datasets. We believe that exploring such structural potentials of DL models is crucial for their further improvements in tabular learning, as the performance gap closes between DL and tree-based methods.

**Limitation and Future work.** An under-explored aspect of this work is the impact of extensive hyper-parameters tuning on tabular learning. In our experiments, we focused our hyper-parameter search to identify, for each method, a set of default values that work well across all datasets. This is because task-level hyper-parameter tuning can be computationally prohibitive (especially for tabular DL) and produces mixed results (Yan et al., 2024; Chen et al., 2023). However, tree-based methods are generally more advantageous than DL ones in terms of computational efficiency, and the former could afford more extensive hyper-parameter search to yield better models. We leave the investigation on the trade-off between computational budget and model performance to future works.

Our proposed framework is a general strategy compatible with recent advancements in tabular DL and beyond. For instance, distillation could be directly combined with transformer architectures tailored for tabular learning (e.g., Somepalli et al., 2021; Chen et al., 2023), or with cross-tabular pre-training (e.g., Zhu et al., 2023; Yan et al., 2024; Ye et al., 2024). In addition, our framework could leverage different data augmentation techniques or language models. We leave these exploration to future works.

## Acknowledgments

This work is supported by Career Development Fund (grant C210812045) from A*STAR Singapore.

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

# Supplementary Material

We provide additional details on the experimental protocol used to compare DisTab with other Tabular learning strategies.

- Appendix A outlines the specific choice of benchmark dataset used in our experiments.

- Appendix B reviews previous methods for tabular prediction (both tree-based and DL-based), including the hyperparameters used in our experiments.

- Appendix C reports in-depth results for the individual datasets, providing further insight on the models performance and comparison.

## A  Dataset Statistics

We use 25 OpenML datasets for our experiments, comprising 8 regression tasks, 10 binary classification tasks, and 7 multi-class classification tasks. Tab. 4 lists the statistics for each dataset, as well as its OpenML ID for reproducibility. The datasets range from 506 samples (boston) to 581012 (covertype), and from 5 columns (blood-transfusion) to 80 (house_prices_nominal).

| Task name | OpenML ID | Number of rows | Number of columns | Task type |
|---|---|---|---|---|
| abalone | 359944 | 4177 | 9 | regression |
| black_friday | 359937 | 166821 | 10 | regression |
| boston | 359950 | 506 | 14 | regression |
| diamonds | 233211 | 53940 | 10 | regression |
| house_prices_nominal | 359951 | 1460 | 80 | regression |
| house_sales | 359949 | 21613 | 22 | regression |
| moneyball | 167210 | 1232 | 15 | regression |
| space_ga | 359933 | 3107 | 7 | regression |
| adult | 7592 | 48842 | 15 | binary |
| bank-marketing | 14965 | 45211 | 17 | binary |
| blood-transfusion | 359955 | 748 | 5 | binary |
| churn | 359968 | 5000 | 21 | binary |
| credit-g | 31 | 1000 | 21 | binary |
| higgs | 146606 | 98050 | 29 | binary |
| kc1 | 3917 | 2109 | 22 | binary |
| kick | 359991 | 72983 | 33 | binary |
| pc4 | 359958 | 1458 | 38 | binary |
| qsar-biodeg | 359956 | 1055 | 42 | binary |
| car | 146821 | 1728 | 7 | multiclass |
| covertype | 7593 | 581012 | 13 | multiclass |
| diabetes130us | 168877 | 101766 | 50 | multiclass |
| okcupid-stem | 359993 | 50789 | 20 | multiclass |
| segment | 146822 | 2310 | 17 | multiclass |
| steel-plates-fault | 168784 | 1941 | 27 | multiclass |
| wine-quality-white | 359974 | 4898 | 12 | multiclass |

Table 4: Table statistics for datasets used in our experiments

## B  Baseline Methods

### B.1  Tree-based models

As tree-based models achieve state-of-the-art performance on tabular tasks (Grinsztajn et al., 2022; Chen et al., 2023; Zhu et al., 2023), we evaluate popular tree-based models comprising XGBoost (Chen & Guestrin, 2016), LightGBM (Ke et al., 2017), CatBoost (Prokhorenkova et al., 2018), and Random Forest (Breiman, 2001). We use the recommended hyperparameters (see Tab. 5, Tab. 6,Tab. 7, Tab. 8), early stopping strategy, and feature pre-processing implemented in AutoGluon (Erickson et al., 2020) 1.0.0 release for each tree-based model, which achieves strong performance on the evaluated datasets.

| Name | Value | Description |
|---|---|---|
| n_estimators | 10000 | Number of boost round |
| max_depth | 6 | Maximum depth of a tree. |
| learning_rate | 0.1 | Learning rate |
| reg_alpha | 0 | $\ell_1$ regularization. |
| reg_lambda | 1 | $\ell_2$ regularization. |
| proc.max_category_levels | 100 | maximum number of allowed levels per categorical feature |
| booster | gbtree | Which booster to use |
| early_stopping_rounds | adaptive | Patience for early stopping adapts to training set size. |

Table 5: Hyperparemeters for XGBoost

| Name | Value | Description |
|---|---|---|
| num_leaves | 30 | Max number of leaves in one tree |
| max_depth | -1 | Max tree depth. -1 denotes unlimited depth. |
| learning_rate | 0.05 | Learning rate |
| n_estimators | 100 | Number of boosting iterations |
| reg_alpha | 0.0 | $\ell_1$ regularization |
| reg_lambda | 0.0 | $\ell_2$ regularization |
| subsample | 1 | Bagging fraction |
| early_stopping_rounds | adaptive | Patience for early stopping adapts to training set size. |

Table 6: Hyperparemeters for LightGBM.

| Name | Value | Description |
|---|---|---|
| learning_rate | 0.05 | Learning rate. |
| random_strengh | 5 | The amount of randomness to use for scoring splits. |
| l2_leaf_reg | 3.0 | $\ell_2$ regularization on leaf node. |
| leaf_estimation_iterations | 1 | Number of iterations for calculating leaf values. |
| iterations | 10000 | Maximum number of trees to be built. |

Table 7: Hyperparemeters for CatBoost

| Name | Value | Description |
|---|---|---|
| n_estimators | 1000 | The number of trees in random forest. |
| max_leaf_nodes | 15000 | Maximum number of leaf nodes. |
| max_features | 0.5 | The number of features to consider when looking for the best split. |
| bootstrap | True | Whether bootstrap samples are used when building trees. |

Table 8: Hyperparemeters for Random forest.

## B.2 DL models

We evaluate a diverse set of DL tabular methods in our experiments, including both MLP-based and transformer-based models. We also consider different pre-training strategies and learning from scratch. We highlight their implementation details below.

**FastAI.** (Howard & Gugger, 2020) We use FastAI as our choice of MLP-based model trained from scratch. It adaptively determines the embedding sizes of input features. We use the AutoGluon implementation with the following hyper-parameters in Tab. 9.

| Name | Value | Description |
|------|-------|-------------|
| layers | [200, 100] | Size of hidden layers for MLP |
| emb_drop | 0.1 | embedding layers dropout |
| ps | 0.1 | linear layers dropout |
| epochs | 30 | number of epochs |
| lr | 1e-2 | learning rate |
| bs | 256 | batch size |

Table 9: Hyperparemeters for FastAI

**Vime.** (Yoon et al., 2020) Vime is a MLP-based tabular learning model. It uses a reconstruction loss for pre-training. Our implementation is based on the repository TabularS3L (`https://github.com/Alcoholrithm/TabularS3L`). The hyper-parameters are listed in Tab. 10.

| Name | Value | Description |
|------|-------|-------------|
| layer count | 3 | Number of hidden layers |
| hidden_dim | 1024 | Dimension of hidden layers |
| pt_epochs | 40 | Training epochs |
| epochs | 100 | Fine-tuning epochs |
| patience | 20 | Early-stopping patience during fine-tuning |
| learning_rate | 1.0e-4 | Learning rate |
| weight_decay | 3e-6 | Weight decay |
| batch size | 128 | Batch size |

Table 10: Hyperparemeters for Vime.

**Scraf.** (Bahri et al., 2022) Scraf is another MLP model with contrastive loss for pre-training. Our implementation is based on TabularS3L, with the hyper-parameters listed in Tab. 11.

| Name | Value | Description |
|------|-------|-------------|
| layer count | 3 | Number of hidden layers |
| hidden_dim | 1024 | Dimension of hidden layers |
| pt_epochs | 40 | Training epochs |
| epochs | 100 | Fine-tuning epochs |
| patience | 20 | Early-stopping patience during fine-tuning |
| learning_rate | 1.0e-4 | Learning rate |
| weight_decay | 3e-6 | Weight decay |
| batch size | 128 | Batch size |

Table 11: Hyperparemeters for Scraf.

**FT-Transformer.** (Gorishniy et al., 2021) FTT is a transformer-based model trained from scratch. We use its implementation from AutoGluon with the following hyper-parameters in Tab. 12.

**XTab.** (Zhu et al., 2023) XTab is a transformer-based model that utilizes multi-task learning for pre-training. It pre-trains a shared transformer across different tabular datasets to learned features generalizable across tables. XTab is based on FT-transformer and shares the same hyper-parameters (Tab. 12). We use the official pre-trained checkpoint of XTab (`https://github.com/BingzhaoZhu/XTab`).

| name | value | Description |
|---|---|---|
| token_dim | 192 | Dimension of input tokens |
| num_blocks | 3 | Number of transformer blocks |
| attention_n_heads | 8 | Number of attention heads |
| head_activation | relu | Activation function of MLP layer performing inference |
| ffn_activation | reglu | Activation function in feed-forward layer of transformer block |
| patience | 20 | Early-stopping patience |
| attention_dropout | 0.2 | Dropout in attention layer |
| ffn_dropout | 0.1 | Dropout in feed-forward layer of transformer block |
| learning_rate | 1.0e-4 | Learning rate |
| weight_decay | 1.0e-5 | Weight decay |
| batch size | 128 | Batch size |

Table 12: Hyperparemeters for FTT and XTab.

**Saint.** (Somepalli et al., 2021) Saint is a transformer-based model with constrastive pre-training. It introduces inter-sample attention block such that different samples within a training batch could attend to one another. We use the official implementation (`https://github.com/somepago/saint`) with the following hyper-parameters.

| Name | Value | Description |
|---|---|---|
| embedding_dim | 32 | Dimension of input tokens |
| attention_n_heads | 8 | Number of attention heads |
| self_head_dim | 16 | Dimension of the heads in Self-Attention block |
| inter_head_dim | 64 | Dimension of the heads in Inter-sample Attention block |
| pt_epochs | 50 | Pre-training epochs |
| epochs | 100 | Fine-tuning epochs |
| attention_dropout | 0.1 | Dropout in attention layer |
| ffn_dropout | 0.8 | dropout rate in feed-forward layer in transformer block |
| learning_rate | 1.0e-4 | learning rate during both pre-traing and fine-tuning |
| weight_decay | 1.0e-2 | Weight decay during both pre-traing and fine-tuning |
| pt_tasks | contrastive, denoising | pre-training objectives |
| batch size | 128 | Batch size |

Table 13: Hyperparemeters for Saint.

**SwitchTab.** (Wu et al., 2024) SwitchTab is a transformer-based model with reconstructive pre-training. During pre-training, the model learns to decouple mutual and salient features for each sample and synthesize corrupted samples by recombining mutual and salient features from different samples. The pre-training objective is thus to recover from the corrupted samples. Our implementation is based on TabularS3L, with hyper-parameters in Tab. 14.

| Name | Value | Description |
|---|---|---|
| token_dim | 192 | Dimension of input tokens |
| num_blocks | 3 | Number of transformer blocks |
| attention_n_heads | 8 | Number of attention heads |
| pt_epochs | 40 | Pre-training epochs |
| epochs | 100 | Fine-tuning epochs |
| patience | 20 | Early-stopping patience during fine-tuning |
| attention_dropout | 0.1 | Dropout in attention layer |
| ffn_dropout | 0.1 | Dropout in feed-forward layer in transformer block |
| learning_rate | 1.0e-4 | Learning rate |
| weight_decay | 3e-6 | Weight decay |
| batch size | 128 | Batch size |

Table 14: Hyperparemeters for SwitchTab.

**DisTab.** Our proposed method uses Llama-3-8B for computing the language-guided embeddings for each column. Specifically, textual information for a column is first embedded by the Llama tokenizer into a sequence of tokens $X \in \mathbb{R}^{n \times d}$, where $n$ is the number of tokens and $d$ the embedding dimension of Llama model. We define $f_{\text{enc}} = f_{\text{proj}} \circ f_{\text{lm}}$, where $f_{\text{lm}} : \mathbb{R}^{n \times d} \to \mathbb{R}^{n \times d}$ outputs the sequence representation $O \in \mathbb{R}^{n \times d}$ from Llama model. $f_{\text{proj}} = \text{MLP}(\text{AvgPool}(O))$, which summarizes the sequence representation into a single vector, followed by a learnable projection MLP.

We list the hyper-parameters used for DisTab below.

| Name | Value | Description |
|---|---|---|
| token_dim | 512 | Dimension of input tokens |
| num_blocks | 3 | Number of transformer blocks |
| attention_n_heads | 8 | Number of attention heads |
| pt_epochs | 30 | Pre-training epochs |
| epochs | 20 | Fine-tuning epochs |
| learning_rate | 1.0e-4 | Learning rate |
| weight_decay | 1e-6 | Weight decay |
| batch size | 1024 | Batch size |
| aug_size | 100000 | Number of synthetic data generated for pre-training |

Table 15: Hyperparemeters for DisTab.

### B.3 Hyper-parameter Tuning

We performed grid search for all methods to determine the key hyper-parameter values that have been reported in the previous section. For each task, we use the validation performance on the first train/test split, as specified by OpenML, to guide the grid search. Validation performance across different tasks is averaged to select the best performing hyper-parameter configuration for each model. We use a single set of hyper-parameters for different tasks, which is a common experiment setting also adopted in previous works (Zhu et al., 2023; Chen et al., 2023; Yan et al., 2024). We stress that robustness to hyper-parameter choice is crucial for tabular models, since they are expected to perform well across diverse datasets. This is especially important for tabular DL methods since hyper-parameter tuning over a large pre-trained model is often prohibitive from the computational standpoint (Yan et al., 2024).

We list the grid search space for each method below.

| Name | Search space |
|------|-------------|
| n_estimators | [100, 1000, 10000] |
| max_depth | [4, 6, 10] |
| learning_rate | [0.01, 0.1, 0.5] |

Table 16: Search Grid for XgBoost

| Name | Search space |
|------|-------------|
| num_leaves | [5, 30, 50] |
| max_depth | [3, 20, -1] |
| learning_rate | [0.01, 0.05, 0.1] |
| n_estimators | [100, 1000, 10000] |

Table 17: Search Grid for LightGBM

| Name | Search space |
|------|-------------|
| learning_rate | [0.01, 0.05, 0.1] |
| random_strengh | [1, 5, 10] |
| l2_leaf_reg | [0.1, 1, 3] |
| iterations | [100, 1000, 10000] |

Table 18: Search Grid for CatBoost

| Name | Grid search space |
|------|-------------------|
| n_estimators | [10, 50, 100, 300, 1000] |
| max_leaf_nodes | [100, 500, 1000, 4000, 15000] |
| max_features | ['sqrt', 0.5, 0.25] |

Table 19: Search Grid for Random Forests

| Name | Search space |
|------|-------------|
| emb_drop | [0.1, 0.2, 0.5] |
| ps | [0.1, 0.2, 0.5] |
| epochs | [30, 50, 100] |
| lr | [0.001, 0.01, 0.1] |

Table 20: Search Grid for FastAI

| Name | Grid search space |
|------|-------------------|
| layer_count | [2, 3, 5] |
| hidden_dim | [256, 512, 1024] |
| epochs | [50, 100] |
| learning_rate | [1e-4, 1e-3, 1e-2] |

Table 21: Search Grid for Vime

| Name | Grid search space |
|------|-------------------|
| layer_count | [2, 3, 5] |
| hidden_dim | [256, 512, 1024] |
| epochs | [50, 100] |
| learning_rate | [1e-4, 1e-3, 1e-2] |

Table 22: Search Grid for Scarf

| Name | Grid search space |
|------|-------------------|
| token_dim | [192, 256, 512] |
| num_blocks | [2, 3, 4] |
| attention_n_heads | [4, 8] |
| learning_rate | [1e-4, 1e-3, 1e-2] |

Table 23: Search Grid for FT-Transformer and XTab

| Name | Grid search space |
|------|-------------------|
| token_dim | [192, 256, 512] |
| num_blocks | [2, 3, 4] |
| attention_n_heads | [4, 8] |
| learning_rate | [1e-4, 1e-3, 1e-2] |

Table 24: Search Grid for SwitchTab

| Name | Grid search space |
|------|-------------------|
| token_dim | [192, 256, 512] |
| num_blocks | [2, 3, 4] |
| attention_n_heads | [4, 8] |
| learning_rate | [1e-4, 1e-3, 1e-2] |

Table 25: Search Grid for DisTab

# C  Further Experiment Results

## C.1  Comparison with Tree-based methods with Average Performance

To compute average performance across different task settings, we transform RMSE metric in regression tasks to $R^2$ score, such that its range and directionality aligns with AUC in binary tasks and accuracy

| | Metric | CAT | RF | GBM | XGB | DisTab |
|---|---|---|---|---|---|---|
| Regression | $R^2$ | 0.81 | 0.784 | 0.795 | 0.795 | 0.809 |
| Binary | AUC | 0.86 | 0.849 | 0.85 | 0.854 | 0.858 |
| Multi-class | ACC | 0.813 | 0.812 | 0.821 | 0.822 | 0.821 |
| All | | 0.831 | 0.818 | 0.824 | 0.826 | 0.832 |

Table 26: Average performance comparison between DisTab and tree-based methods

for multi-class tasks. Specifically, all three metrics share the range of $[0,1]$ and larger values imply better performance.

We report the average task performance in Tab. 26. The results are consistent with those reported in Tab. 2. In particular, DisTab matches the performance of CatBoost overall and outperform the others. In the multi-class setting, DisTab noticeably outperforms CatBoost and matches the performance of LightGBM and XgBoost.

## C.2   Attention Activation Visualization

We visualize the average attention activation of DisTab v.s. FT on moneyball and steel-plates-fault datasets below. The results suggest while both model variants learned to identify similar key predictive features, language guidance provides slightly different feature mixing compared to FT. In both datasets, language guidance yielded better test performance.

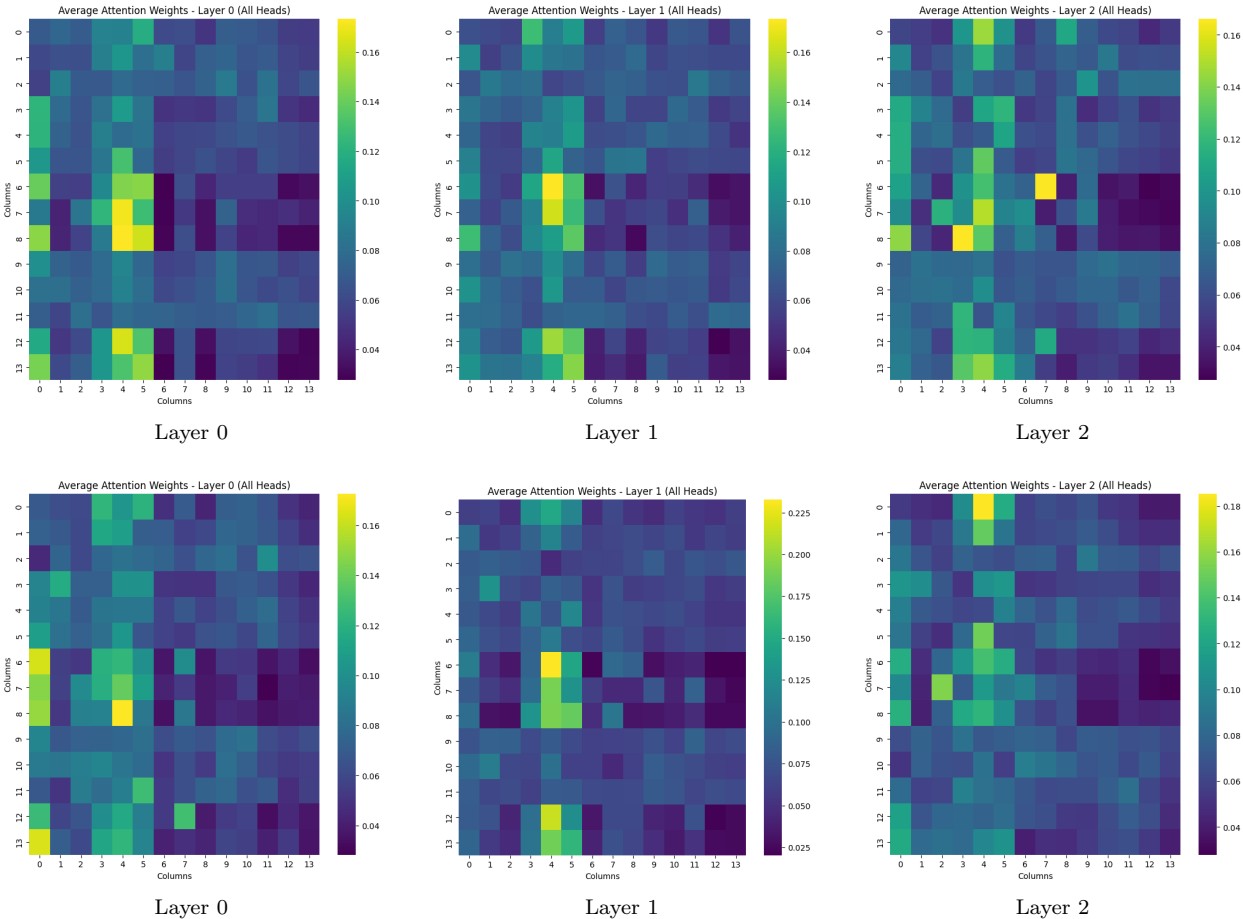

Figure 6: Average attention activation on FT vs. DisTab on moneyball.

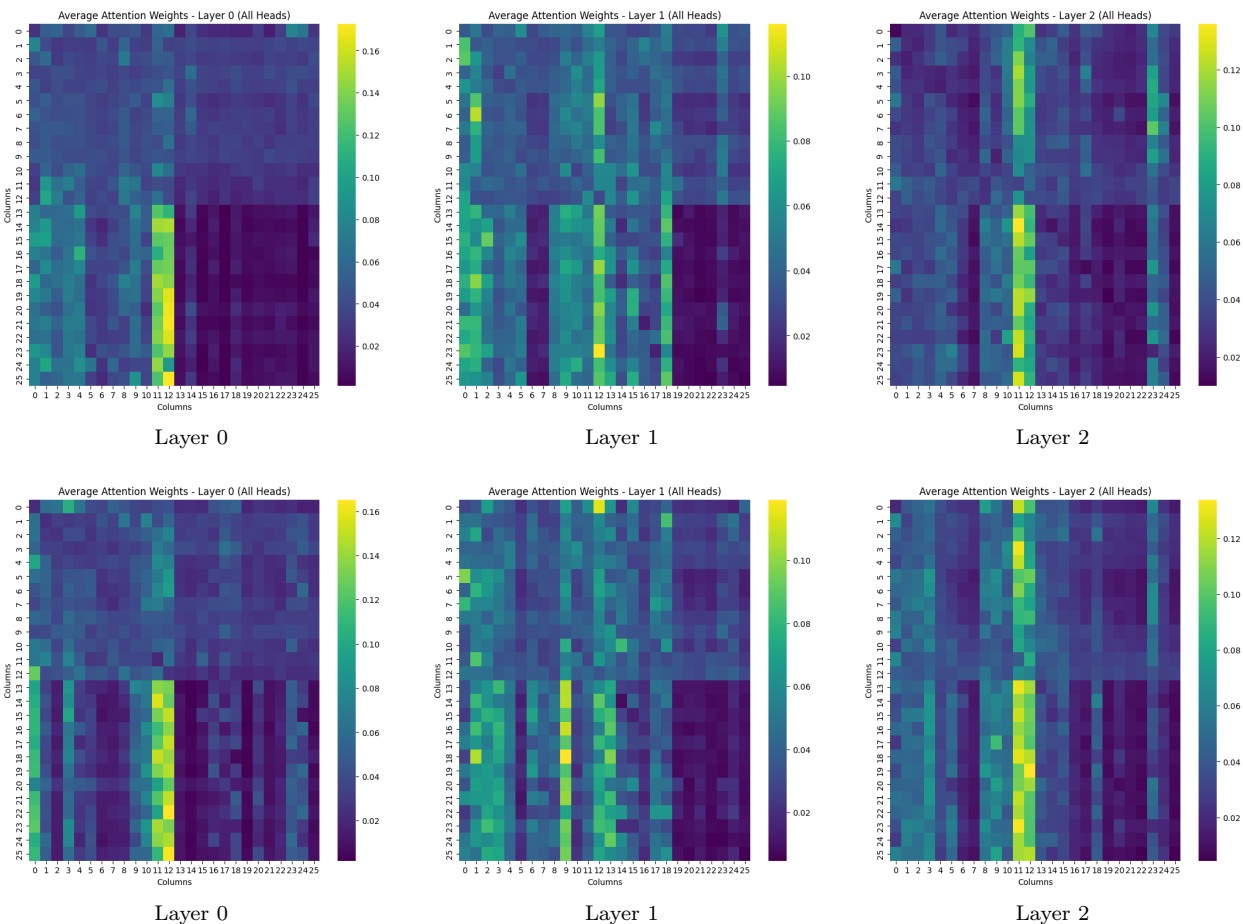

Figure 7: Average attention activation on FT vs. DisTab on steel-plates-fault.

## C.3 Individual Task Performance

In Tab. 27, we report performance of all evaluated methods for each individual tasks. Each task contains 5 data splits and the average performance over the splits is reported. The evaluated tasks are discussed in Appendix A and the evaluated methods in Appendix B.

| Dataset | Metric | Tree-based | | | | MLP-based models | | | | | Transformer-based models | | | |
|---|---|---|---|---|---|---|---|---|---|---|---|---|---|---|
| | | CAT | RF | GBM | XGB | FastAI | Vime | Scraf | FTT | XTab | Saint | TP-BERTa | SwitchTab | Ours |
| abalone | RMSE | 2.197 | 2.184 | 2.203 | 2.215 | 2.133 | 2.165 | 2.18 | 2.134 | 2.18 | 2.152 | 2.288 | 2.203 | 2.172 |
| black_friday | RMSE | 3456.0 | 3499.1 | 3446.5 | 3452.3 | 3592.8 | 3545.3 | 3519.1 | 3522.0 | 3521.3 | 3524.0 | 3515.2 | 3534.5 | 3445.0 |
| boston | RMSE | 2.672 | 3.137 | 3.422 | 3.266 | 3.987 | 3.349 | 3.023 | 3.513 | 3.351 | 3.481 | 4.293 | 3.034 | 2.439 |
| diamonds | RMSE | 512.7 | 549.7 | 525.2 | 539.1 | 553.7 | 550.7 | 606.3 | 516.2 | 519.9 | 529.5 | 633.7 | 538.7 | 515.9 |
| house_prices | RMSE | 21820.7 | 24814.9 | 25297.9 | 24092.5 | 23953.8 | 26589.2 | 26729.2 | 22717.1 | 22595.1 | 25608.1 | 26088.8 | 25484.5 | 23204.2 |
| house_sales | RMSE | 106163.1 | 121409.6 | 111775.9 | 114148.0 | 112693.5 | 121612.1 | 120026.5 | 111015.6 | 110752.9 | 117957.4 | 116707.8 | 117931.3 | 107590.2 |
| moneyball | RMSE | 22.86 | 24.25 | 23.99 | 24.24 | 22.33 | 23.59 | 24.69 | 22.01 | 21.86 | 23.85 | 27.22 | 23.4 | 21.85 |
| space_ga | RMSE | 0.1006 | 0.1098 | 0.1034 | 0.1053 | 0.1004 | 0.105 | 0.1025 | 0.1017 | 0.105 | 0.1012 | 0.1298 | 0.1028 | 0.1063 |
| adult | AUC | 0.9287 | 0.9098 | 0.9287 | 0.9286 | 0.9106 | 0.9119 | 0.9115 | 0.9159 | 0.9157 | 0.9217 | 0.9290 | 0.9110 | 0.9301 |
| bank-marketing | AUC | 0.9383 | 0.9306 | 0.9377 | 0.9372 | 0.9341 | 0.9338 | 0.9312 | 0.9381 | 0.9380 | 0.9378 | 0.9436 | 0.9212 | 0.9403 |
| blood-transfusion | AUC | 0.7557 | 0.7317 | 0.7328 | 0.7378 | 0.7443 | 0.7465 | 0.7625 | 0.7625 | 0.7595 | 0.7465 | 0.7389 | 0.7539 | 0.7416 |
| churn | AUC | 0.9195 | 0.9110 | 0.9114 | 0.9204 | 0.9059 | 0.9024 | 0.9079 | 0.9119 | 0.9202 | 0.9088 | 0.8773 | 0.9176 | 0.9194 |
| credit-g | AUC | 0.7706 | 0.7757 | 0.7540 | 0.7508 | 0.7407 | 0.7630 | 0.7750 | 0.7405 | 0.7460 | 0.7896 | 0.7534 | 0.7405 | 0.7729 |
| higgs | AUC | 0.8131 | 0.8031 | 0.8097 | 0.8080 | 0.8133 | 0.7873 | 0.7733 | 0.8164 | 0.8154 | 0.7993 | 0.7965 | 0.8118 | 0.8210 |
| kc1 | AUC | 0.8131 | 0.8174 | 0.7886 | 0.7984 | 0.8061 | 0.8037 | 0.7931 | 0.7934 | 0.8000 | 0.7929 | 0.7868 | 0.7946 | 0.8066 |
| kick | AUC | 0.7826 | 0.7612 | 0.7676 | 0.7840 | 0.7628 | 0.7808 | 0.7682 | 0.7735 | 0.7741 | 0.7774 | 0.7676 | 0.7484 | 0.7732 |
| pc4 | AUC | 0.9495 | 0.9446 | 0.9478 | 0.9494 | 0.9262 | 0.9267 | 0.9266 | 0.9445 | 0.9496 | 0.9484 | 0.8645 | 0.9339 | 0.9503 |
| qsar-biodeg | AUC | 0.9262 | 0.9188 | 0.9208 | 0.9245 | 0.9294 | 0.9310 | 0.9187 | 0.9247 | 0.9287 | 0.9236 | 0.9151 | 0.9189 | 0.9224 |
| car | Acc | 98.38 | 97.92 | 98.96 | 98.27 | 98.38 | 99.19 | 99.19 | 99.65 | 99.08 | 97.57 | 99.19 | 99.08 | 99.54 |
| covertype | Acc | 94.14 | 93.18 | 97.18 | 97.00 | 91.33 | 96.36 | 96.93 | 97.28 | 97.16 | 90.79 | 96.69 | 96.39 | 95.73 |
| diabetes130us | Acc | 60.98 | 60.61 | 60.45 | 61.01 | 59.28 | 60.28 | 60.53 | 61.04 | 60.95 | 60.08 | 60.53 | 59.41 | 61.42 |
| okcupid-stem | Acc | 75.54 | 75.61 | 75.95 | 76.24 | 74.82 | 74.87 | 75.06 | 75.27 | 75.06 | 75.11 | 75.94 | 72.17 | 76.20 |
| segment | Acc | 92.73 | 94.37 | 93.85 | 93.68 | 91.86 | 91.77 | 92.64 | 92.90 | 93.07 | 93.33 | 92.29 | 92.55 | 94.46 |
| steel-plates-fault | Acc | 79.71 | 79.71 | 81.36 | 80.95 | 75.59 | 76.42 | 77.34 | 78.99 | 75.18 | 77.55 | 72.61 | 77.55 | 81.05 |
| wine-quality-white | Acc | 67.88 | 69.06 | 67.22 | 67.96 | 58.12 | 65.10 | 64.49 | 60.78 | 58.82 | 62.37 | 65.43 | 63.88 | 66.12 |

Table 27: Average task performance over 5 data split for baseline methods and DisTab. The results are used to compute the win matrices and average ranking in Sec. 4 in the main text.

