# OpenReview forum: "Deep Tabular Learning via Distillation and Language Guidance"
_TMLR — Accepted by TMLR_

### Review · Reviewer_8cMY · 2024-09-23

**Summary Of Contributions:**

This paper discusses deep learning (DL) methods for tabular data, comparing to non-DL (tree-based) approaches that are in general superior to DL approaches in tasks using tabular data. Pointing out that existing DL approaches miss potential of semantic context in tabular data such as column headers and text descriptions, the authors introduce language-guided categorical / numerical embeddings to their DL approach. Besides, they leverage a pretrained tree-based method to generate additional training datasets for pretraining DL models. With 25 tabular datasets in three different types of tasks (regression, binary classification, multi-class classification), they demonstrated that the proposed method 1) outperforms DL-based baselines and 2) either outperforms or matches tree-based baselines, according to average ranks and win matrices. Their ablation study suggests that the combination pre-training, language-guided embeddings, and fine-tuning is the best configuration for their proposed method.

**Audience:**

Yes

**Claims And Evidence:**

No

**Requested Changes:**

The reviewer requests changes in the manuscript by addressing the following comments / questions in addition to weaknesses shared in "Strengths And Weaknesses"

# Comments
- The authors claim this work's contributions are three-fold, but 1) cannot be a contribution without 2), i.e., the contribution of this work are two-fold: 1) + 2) and 3).
- Proposed model implementation is not clear enough, from "Model Architecture" in Section 3.3 and Table 15. It would be helpful if the authors offers a figure of the end-to-end network architecture.


# Questions
> One added benefit of distillation, compared to existing approaches, is that the pre-trained model can be directly used for inference without further fine-tuning.

The reviewer does not understand why the authors claim this as a benefit. Table 3 in this work suggests just pre-training ("Distil" in the table) is not enough.

---

Given a task, the embeddings are fixed i.e. this is a sort of a bias term, and the reviewer feels the look-up embeddings and random feature embeddings in Figs. 1 and 2 are enough.
What are models supposed to learn about given Language-Guided Embeddings?

**Strengths And Weaknesses:**

# Strengths

- This paper offers detailed explanations about hyperparameters and stats of datasets used in this study, which are important for reproducibility in this work.
- It seems to the reviewer that using a pretrained non-DL model for generating additional datasets makes sense for DL approaches for tasks using tabular datasets
- Its presentation of results, especially Table + Win matrix seems like a nice presentation that helps readers compare different methods.

# Weaknesses

A critical weakness of this study is that there are no justifications of hyperparameter choices for baselines, and it is unclear how the hyperparameters are determined for baseline and propsoed methods. Appendix B says that this work used the default hyperparameters, and the reviewer believes that it is unfair to discuss the performance of baseline methods vs. the proposed method in this manner. It also makes the reviewer feel that it is difficult to see if the reported empirical results support claims in this paper.

---

The reviewer also thinks that motivation of this study is not strong and needs more clarifications;
> To address these limitations, we introduce DisTab, a new tabular DL framework aimed at bridging the gap with tree-based approaches. DisTab leverages knowledge distillation (Hinton et al., 2015) for pre-training, directly employing a suitable tree-based model as the teacher. This enables our model to emulate the favorable inductive biases inherent in tree-based approaches (Grinsztajn et al., 2022), effectively closing the performance disparity between tabular DL and GBDTs.

Why do DL methods need to improve tree-based methods? In other words, why does the solution have to be DL-based?

---

One of the key acceptance criteria at TMLR is "Are the claims made in the submission supported by accurate, convincing and clear evidence?"
From this perspective, the reviewer finds the following claims do not meet the criteria

Claim
> DisTab showcases how tabular DL models can capitalize on the flexibility of neural architectures (specifically transformer architectures in our case) to organically incorporate multiple information sources without need for ad-hoc model design.

The reviewer couldn't find the strong evidence to support the claim. Also, it is unclear what flexibility of neural architectures means in this context.

---
Claim
> We show that this choice enhances the conventional tabular embeddings with semantic context to improve generalization performance. This integration serves two purposes: from the practical perspective, it enables DisTab to achieve state-of-the-art performance on tabular
data. From the methodological perspective, it showcases the benefits of tabular transformers as a natural architecture to integrate different format of information in contrast to the effective yet rigid structure of tree-based methods (for which language integration would be less straightforward).

The reviewer couldn't find clear, strong evidence that supports the need of LM paths used in Figs. 1 and 2. Language-guided embeddings used in the ablation study seem like a combination of existing embeddings (top paths in Figs. 1 and 2) and LM-based embeddings, but contribution of the latter as a single component is not clear.


## Minor

> A critical limitation of these models is however the ability to deal with large scale vectorial data (such as the
representation from a visual or language model). This makes tabular DL a potential competitor in settings
where input data includes diverse modalities including vectorial data (e.g. language-based information in
this work).

The reviewer cannot agree to the above statement. This paper is about tabular learning, which should be focused on learning from tabular data. On top of that, the authors do not provide any concrete tasks that mainly use tabular data where other modalities (e.g., text, vision) are critical. For these reason, the reviewer does not think that it is a critical limitation of GBDT models.

## Typos
- "as it the case" ???
- There are many lines that miss a prefix "Eq." e.g., "the latent dimension d in (3) increases", "The embedding process modeled in (2) is", "from (1) and (2).", and more.
- "in teh kernel" -> "in the kernel"

---

> ### Author Response · Authors · 2024-09-26
> **Reply 1 of 3**
>
> We thank the reviewer for the feedbacks. Please see our responses below.
>
> **1\. Hyper-parameters choices**
>
> “Default“ hyper-parameters refer to a fixed set of hyper-parameters per method applied across all tabular datasets. The term does not denote an arbitrary choice of hyperparameters but rather a fixed set previously selected according to a suitable heuristic (e.g. using Autogluon parameters for tree-based methods, as discussed in [1]). We stress that this is a standard evaluation setting in previous works (e.g, [1, 2]). We adopted this setting and kept the term “default hyper-parameters” for consistency. We will clarify this in the main paper.
>
> For tree-based methods, [1] reported model performance under both default and hyper-parameter search (HPO) settings. The results in [1] show that models perform comparably in both settings. For completeness, in Table R1 below, we report the performance of tree-based methods obtained in [1] and our reported tree-based results. We observe that our reported results are essentially equivalent to those obtained in [1].
>
> | Task               | Metric   |   XGB (Ours) |      XGB[1]       |   XGB HPO[1]      |   LGBM (Ours) |      LGBM[1]      |   LGBM HPO[1]     |   CAT (Ours) |      CAT[1]       |   CAT HPO[1]      |
> |:-------------------|:--------:|-----------:|:------------------:|:------------------:|:------------:|:------------------:|:------------------:|:-----------:|:------------------:|:------------------:|
> | abalone            | RMSE     |     2.2147 |  2.2099 (▲0.22%)   |  2.1927 (▲0.99%)   |     2.2028   |  2.2091 (▼0.29%)   |  2.2062 (▼0.15%)   |    2.1972   |  2.1944 (▲0.13%)   |  2.2103 (▼0.6%)    |
> | black_friday       | RMSE     |  3452.2967 | 3459.81 (▼0.22%)   | 3452.056 (▲0.01%)  |   3446.497   | 3452.612 (▼0.18%)  | 3452.454 (▼0.17%)  | 3456.0489  | 3462.058 (▼0.17%)  | 3463.792 (▼0.22%)  |
> | diamonds           | RMSE     |   539.1241 | 540.029 (▼0.17%)   | 534.047 (▲0.94%)   |    525.209   | 525.7748 (▼0.11%)  | 521.6772 (▲0.67%)  | 512.7189   | 514.932 (▼0.43%)   | 517.6136 (▼0.95%)  |
> | adult              | AUC      |     0.9286 | 0.9282 (▼0.04%)    | 0.9288 (▲0.02%)    |     0.9287   | 0.9286 (▼0.01%)    | 0.928 (▼0.08%)     | 0.9287     | 0.9287 (▼0.0%)     | 0.929 (▲0.03%)     |
> | bank-marketing     | AUC      |     0.9372 | 0.9364 (▼0.09%)    | 0.9364 (▼0.09%)    |     0.9377   | 0.9372 (▼0.05%)    | 0.9385 (▲0.09%)    | 0.9383     | 0.9387 (▲0.04%)    | 0.9388 (▲0.05%)    |
> | pc4                | AUC      |     0.9494 | 0.9478 (▼0.17%)    | 0.9366 (▼1.35%)    |     0.9478   | 0.9507 (▲0.31%)    | 0.9437 (▼0.43%)    | 0.9495     | 0.9519 (▲0.25%)    | 0.9425 (▼0.74%)    |
> | diabetes130us      | log loss |     0.8422 | 0.8421 (▲0.01%)    | 0.8357 (▲0.77%)    |     0.855    | 0.8563 (▼0.15%)    | 0.8499 (▲0.6%)     | 0.8357     | 0.836 (▼0.04%)     | 0.8355 (▲0.02%)    |
> | okcupid-stem       | log loss |     0.5616 | 0.5700 (▼1.5%)     | 0.5663 (▼0.84%)    |     0.5665   | 0.5722 (▼1.01%)    | 0.5701 (▼0.64%)    | 0.558      | 0.564 (▼1.08%)     | 0.5637 (▼1.02%)    |
> | steel-plates-fault | log loss |     0.4905 | 0.4905 (▼0.0%)     | 0.4937 (▼0.65%)    |     0.4978   | 0.4978 (▼0.0%)     | 0.4912 (▲1.33%)    | 0.4777     | 0.4777 (▼0.0%)     | 0.4834 (▼1.19%)    |
>
> **Table R1**. Comparing [1]'s default and HPO tree-based results with our tree-based ones. Upward arrows (▲) indicate better performance with respect to ours, while downward arrows (▼) indicate worse performance.
>
> For clarity, Table R2 summarizes Table R1 by reporting average rank and the average performance percentage difference across all tasks, using our results as the reference. We see that our results on average are within ~1% of those from [1].
>
> |                    |   XGB   |   LGBM   |   CAT   |
> |:-------------------|:-------:|:-------:|:-------:|
> | Ours (%)           |   0.0   |   0.0   |   0.0   |
> | default [1] (%)     |  ▼0.22  |  ▼0.17  |  ▼0.14  |
> | HPO [1] (%)         |  ▼0.02  |  ▲0.13  |  ▼0.51  |
>
> **Table R2**. Average performance percentage difference over all tasks. Our results used as reference.
>
> For DL-based models, we conducted a grid search for each method (including ours) over fixed validation sets to determine default hyper-parameters.
>
> [1] Xtab: Cross-table pretraining for tabular transformers, Zhu et al.
>
> [2] Trompt: Towards a better deep neural network for tabular data, Chen et al.

---

> ### Author Response · Authors · 2024-09-26
> **Reply 2 of 3**
>
> **2\. Clarifications on motivation. “Why do DL methods need to improve tree-based methods? In other words, why does the solution have to be DL-based?”**
>
> The reviewer’s question is addressed on multiple occasions in our paper. In particular, Paragraph 5 of Section 1:
>
> > *Despite these challenges, there exist compelling motivations for leveraging DL models for tabular learning. These include the capacity of DL models to generate expressive tabular representations for downstream tasks (Grinsztajn et al., 2022) and their potential for knowledge transfer across different tabular datasets (Wang & Sun, 2022; Hollmann et al., 2022). Moreover, DL allows for more efficient integration of different input modalities or information sources to enhance model capabilities (as is the case for vision-language models (Alayrac et al., 2022; Zhang et al., 2021)).*
>
> We further elaborate on DL's potential advantages over tree-based methods:
>
> 1) **Flexibility**: High-dimensionality and complex feature interactions typically pose significant challenges to tree-based methods. In contrast, DL models are well documented in the machine learning literature for learning complex feature interactions. It is therefore scientifically relevant to investigate strategies to exploit DL’s such ability in tabular settings. Please see also our reply to Question 3 below.
>
> 2) **Transferability**: DL is known to be well-suited to transfer knowledge across related scenarios, such as in multi-task, meta-learning, transfer learning, continual learning, to name but a few. In contrast, it is unclear how tree-based methods could be directly used for knowledge transfer. Given the ubiquity of tabular data, having a system capable of transferring knowledge acquired on past data to new applications is clearly appealing, as explored in [4, 5]. But to achieve this, DL methods first need to achieve comparable performance to tree-based methods. Therefore, it is scientifically relevant to investigate DL vs. tree-based methods.
>
> 3) **Integration**: The modularity of DL is particularly suited to combine multiple sources of information (textual, visual, numerical, etc.). This means that DL methods could more easily integrate different forms of information available in tabular settings (headers, descriptions of the task, etc.). In contrast, it is unclear how tree-based methods should make use of such information. In this paper, we focus on one of the simplest forms of additional information: headers and tags. We show a natural method to integrate side information into a tabular DL model and report the benefits of such an approach. Again, this shows that it is scientifically relevant to ask whether tabular problems could be effectively tackled by DL models.
>
> Finally, we highlight that the question of whether/how DL could match or outperform tree-based methods is of significant relevance and remains unresolved within the tabular learning research community. This importance is evidenced by numerous prior studies [e.g., 6, 7, 8] that have specifically explored this question.
>
> We will revise our introduction to further strengthen the motivation, with the above points.
>
> [4] Transtab: Learning transferable tabular transformers across tables, Wang et al.
>
> [5] Tabpfn: A transformer that solves small tabular classification problems in a second, Hollmann et al.
>
> [6] Why do tree-based models still outperform deep learning on typical tabular data, Grinsztajn et al.
>
> [7] Tabular Data: Deep learning is not all you need, Shwartz-Ziv et al.
>
> [8] Trompt: Towards a better deep neural network for tabular data, Chen et al.
>
> ---
> **3\. Supporting claim. “Capitalizing on the flexibility of neural architectures”**
>
> We clarify that “flexibility of neural architecture“ refers to the DL (in particular the transformer) architecture ability to easily integrate different information sources or representations. This is well-established in the application of transformer models (e.g., language-vision models). Concretely in our case, our model integrates header information with cell values into the model to improve generalization performance. The performance improvement is supported by the comparison between FT vs DisTab in Table 3 and Figure 6, where the only difference is that FT doesn't incorporate header information. In Table 3, the improvement is particularly significant in multi-class classification tasks, with an average rank improvement of 0.79 from FT to DisTab. In Figure 6, DisTab outperforms FT in 18/25 tasks.
>
> In contrast, we emphasize that it is neither trivial nor clear how tree-based methods could leverage such unstructured header information. For example, the naive implementation of embedding table headers as vectors and using the vectors as additional columns turns out to be detrimental to overall performance.
>
> Please refer also to our reply to Question 2 above.

---

> > ### Author Response · Authors · 2024-09-26
> > **Reply 3 of 3**
> >
> > **4\. “Supports the need of LM paths”**
> >
> > In Table 3 and Figure 6, FT vs DisTab shows the benefits of integrating LM-based embeddings, which improves generalization performance in all task settings, as detailed in our reply to Question 3 above.
> >
> >
> > We designed the LM-embedding to augment the existing embeddings, not as a replacement. For example, random embeddings for numerical data aim to capture other rows with similar column values, but may not follow a linear relationship, as the scaling LM embedding is trying to model.
> >
> > ----
> > **5\. Number of contributions.**
> >
> > We sincerely do not understand the reviewer’s point. Designing a framework is different from evaluating the performance of one of its instantiations.
> >
> > We are confident that the DisTab framework is of interest per se, since it provides a general strategy to design new tabular DL methods. This is a separate contribution from experiments since the abstract model can be used by other researchers for further development. In this sense, experiments are solely meant to support the efficacy of such a strategy and are therefore a separate contribution.
> >
> > That said, we agree with the reviewer that this is relatively minor discussion. The list of contributions is primarily meant as a guide for readers to conceptually navigate a paper.
> >
> > ---
> > **6\. Limitation of GBDT models**
> >
> > Table 3 and Figure 6 explicitly shows that incorporating header information can noticeably improve model performance. In addition, header information is readily available for many tables and therefore it is a missed opportunity not to leverage it for learning (see also [9] for the use of header information in tabular learning). Our statement on the limitation of GBDT refers to its difficulty in handling header information as unstructured input.
> >
> > Our experiments also focus significantly on whether having table headers as a different source/modality of input to tabular models improves generalization performance.
> >
> > [9] Learning transferable tabular transformers across tables, Wang et al.
> >
> > ---
> >
> > **7\. End-to-end network diagram.**
> >
> > We will include an end-to-end network architecture figure in the revised manuscript. As per TMLR guidance, we will update the manuscript after all reviews are posted.
> >
> > ---
> >
> > **8\. Missing Eq. prefix for math references**
> >
> > The reviewer appears to be unfamiliar with the standard IEEE editorial style: “*Use the word ‘Equation’ at the start of a sentence, but in text, just use the number [e.g., in (1)].*” ([IEEE style manual](https://journals.ieeeauthorcenter.ieee.org/wp-content/uploads/sites/7/IEEE-Editorial-Style-Manual-for-Authors.pdf#page=23.66), Page 24 Line 1). Numerous works using this well-established style can be found in the machine learning literature. For instance, [10], a seminal work in machine learning, adheres to this convention (e.g., Page 16, Lemma 4 and onwards).
> >
> > We thank the reviewer for the other typos and will correct them.
> >
> > [10] "On the mathematical foundations of learning." Cucker, Felipe, and Steve Smale.
> >
> > ---
> >
> > **9\. Clarify the benefit of “pre-trained model can be directly used for inference without further fine-tuning”**
> >
> > Other pre-training approaches use surrogate objectives distinct from the desired classification/regression objectives. Thus, the qualities of such pre-trained models cannot be directly evaluated with respect to the target objective. In contrast, our pre-trained model can be directly evaluated on the target objectives by design, leading to easier pre-training tuning and model selection via validation data.
> >
> > We will expand the corresponding discussion in the revised manuscript.
> >
> > ---
> >
> > **10\. Clarifying on language-guided embedding**
> >
> > We first clarify that the language-guided embeddings are not fixed. As described in more detail in Appendix B2, the language embedding has a learnable projection layer. We will make this clearer in the main paper when describing the model architecture.
> >
> > Table 3 and Fig 6 show that incorporating header information improves model performance noticeably (compare FT vs DisTab). This suggests that having only “the look-up embeddings and random feature embeddings” is insufficient.
> >
> > In contrast, since a key mechanism in transformers is attention and feature mixing, we hypothesized that by providing (con)textual information on what each column is about, the model could better relate relevant columns together to produce more robust features, instead of having to learn such relations from column values alone. The improved model performance observed in our experiments supports this intuition.

---

> ### Comment · Reviewer_8cMY · 2024-09-27
> **Reviewer response (1/2)**
>
> The reviewer thanks the authors for the detailed responses and clarifications.
>
> ---
> ## Re: 1. Hyper-parameters choices
>
> The reviewer finds the response not convincing since the experimental settings are different.
> - The reviewer would not consider evaluation setting used in two previous papers as a standard or a strong justification either. In this case, if the authors believe the setting is reasonable, they should explain why it's reasonable in their words.
> - Table R1 misses some baseline (e.g., RF) and majority of the 25 tasks in the main paper.
> - The authors argue that their reported results are "essentially equivalent" to those in [1], referring to the differences between their numbers and the numbers reported in [1]. But, this paper uses win matrices and average ranks as evaluation metrics, which are relative comparisons could significantly change even with $\pm 0.01$.
>
> > For DL-based models, we conducted a grid search for each method (including ours) over fixed validation sets to determine default hyper-parameters.
>
> - There should be more details about the grid search for DL-based models. What parameters are tuned via the grid search? By fixed validation sets what do the authors mean? It is confusing especially because the evaluation section implies that this work used 5-fold cross validation
> >  For each task, every model is trained and evaluated using the same 5 splits, and we use the
> average performance over the 5 splits as a model’s task performance.
>
> In short, the reviewer highly suggests that the authors rerun experiments, tuning hyperparameters for all baselines, and describe the hyperparameter tuning process in the paper.
>
> ---
>
> ## Re: 2. Clarifications on motivation. “Why do DL methods need to improve tree-based methods? In other words, why does the solution have to be DL-based?”
>
> The paragraph 5 of Section 1 in the current manuscript is not detailed enough to explain the motivation. If the revision stresses the points the authors clarified, the motivation of this study will look stronger.
>
> ---
>
> ## Re: 3. Supporting claim. “Capitalizing on the flexibility of neural architectures”
>
> The reviewer thanks the authors for the clarifications. Those address the concerns except one involving experimental results due to the remaining concerns in "1. Hyper-parameters choices"
>
> ---
>
> ## Re: 4. “Supports the need of LM paths”
>
> > We designed the LM-embedding to augment the existing embeddings, not as a replacement. For example, random embeddings for numerical data aim to capture other rows with similar column values, but may not follow a linear relationship, as the scaling LM embedding is trying to model.
>
> This is the point. The reviewer cannot find the results that show the contribution of 1) LM embedding w.r.t. 2) the existing embeddings.The paper shows the results of 1) + 2), but there is no comparison 1) vs. 2) vs. 1) + 2). Especially, 1) + 2) vs. 2) is critical to support the claims regarding the language-guided embeddings.
>
> ---
>
> ## Re: 5. Number of contributions.
>
> The reviewer argues that proposing ideas itself should not be considered as a contribution. There are a few reasons:
> 1. Proposing an idea is important for research communities, but without discussing whether or not the idea actually works well, we cannot assess the idea except its novelty. I.e., not all proposals are significant contributions to the communities. In an empirical study like this work, running experiments is a method to discuss whether the idea is great.
> 2. The novelty is not important for TMLR as much as for other conferences / journals. TMLR is more focused on assessing the technical soundness as well as the clarity of the narrative and arguments presented in the work.
> 3. While the authors claim "the abstract model can be used by other researchers for further development", research communities would not use others' models that have not been fairly assessed and/or open-sourced, neither of which was validated/confirmed in this work at this moment.
>
> ---

---

> > ### Comment · Reviewer_8cMY · 2024-09-27
> > **Reviewer response (2/2)**
> >
> > ## Re: 6. Limitation of GBDT models
> > The response doesn't address the reviewer's concerns in the initial review. The reviewer checked [9], but couldn't find evidence that show the sole contribution of the header information i.e., there is no comparison between the proposed method vs. the proposed method without the header embeddings. For this reason, the reviewer repeats that the authors have not provided any concrete tasks yet that mainly use tabular data where other modalities (e.g., text, vision) are critical and does not think that lack of multimodal support is a critical limitation of GBDT models in context of tabular learning.
> >
> > The response from the authors also made the reviewer wonder why [9] was not used as a baseline in this study. [9] already uses a similar approach, but is just referenced in this paper.
> >
> > ---
> >
> > ## Re: 7. End-to-end network diagram.
> >
> > It should be helpful. The reviewer will be waiting for the update.
> >
> > ---
> >
> > ## Re: 8. Missing Eq. prefix for math references
> >
> > Though the reviewer is not sure why the authors referred to IEEE style at TMLR, the reviewer just called out in case it was not intentional.
> >
> > ---
> >
> > ## Re: 9. Clarify the benefit of “pre-trained model can be directly used for inference without further fine-tuning”
> >
> > The reviewer now understands some points in the claim.
> >
> > > In contrast, our pre-trained model can be directly evaluated on the target objectives by design, leading to easier pre-training tuning and model selection via validation data.
> >
> > The reviewer thinks that to support the claim, Figure 6 should be expanded with baselines. Currently "Distil" is compared only against variants of the proposed model, and the reviewer cannot see if the pre-trained model can work as claimed in the response.
> >
> > ---
> >
> > ## Re: 10. Clarifying on language-guided embedding
> >
> > Sorry for the confusion. The reviewer should have said text embeddings (bottom path in Figs. 3 and 6), which are fixed.
> > Thus, comparing DisTab with FT in Table 3 does not show contribution of the text embeddings. Instead, it shows contribution of the combination of text embedding AND an existing approach, that is learnable look-up / random feature embeddings.
> >
> > ---
> >
> > The reviewer wants to see other reviewers' comments and recommends that the authors' next response follow reviews from the other reviewers. The reviewer's intention in this early follow-up is to give the authors to rerun experiments.

---

> > > ### Author Response · Authors · 2024-09-27
> > > **Clarifications related to LM embeddings**
> > >
> > > We thank the reviewer for the discussion. We wish to first clarify on the remaining concerns related to LM embeddings/path, that is Re:4, 6 and 10.
> > >
> > > In our ablation study, FT denotes a variant of our model that uses pre-training, "existing embeddings" without the LM path, and fine-tuning, while DisTab uses pre-training, "existing embeddings" + LM path, and fine-tuning. This is precisely the "1+2 vs. 1" that the reviewer requested, where 1 is the "existing embeddings" and 2 is the LM path.
> > >
> > > We will revise the description of FT to make this clearer.
> > >
> > > In our previous response, we noted that DisTab noticeably outperforms FT in Table 3 and Figure 6, where the only difference is the incorporation of LM paths. This shows that adding header information helps to improve generalization performance.
> > >
> > > The LM path is represented by $f_{enc}$ as denoted in Eq. 1 in our main text. The specific definition of $f_{enc}$ was detailed in Appendix B2 (Page 18) and it contains a learnable projection layer. Thus the LM path is also not fixed. As noted in our previous response, we will move the definition of $f_{enc}$ to the main text to make this clearer.
> > >
> > > We hope that the above points clarify remaining concerns over LM embedding/path.

---

> > > > ### Comment · Action_Editor_sMx3 · 2024-10-04
> > > > **Format of references to equations**
> > > >
> > > > I appreciate the authors' and reviewers' active discussions. Regarding the format issue of references to equations in **8. Missing Eq. prefix for math references**, TMLR does not have format rules. We let the authors use the format they like as long as it is reasonable.
> > > >
> > > > Best,
> > > > AE

---

> ### Author Response · Authors · 2024-10-05
> **Further Clarifications 1 of 2**
>
> We provide below additional clarifications to the reviewer's response to our initial reply.
>
> **1\. Hyper-parameters choices**
>
> We note that our experiment settings closely follow that of [1], with average model performance over 5 folds as task performance. We also used a subset of the tasks used in [1]. Thus we politely disagree with the comment that “the experimental settings are different”. As shown in the comparison in our initial response, our reported tree-based results closely match those from [1].
>
> In Table R3 below, we compare DisTab’s results to tree-based methods based on the grid-searched hyper-parameters. The results show that RF, XGB, XGB and CatBoost retain the levels of performance similar to the default setting. The results are consistent with what was observed in [1].
>
> | Task Type   |   CAT|   RF |   GBM |   XGB |   Ours |
> |:------------|:---------:|:--------:|:------:|:------:|:-------:|
> | Regression  |     1.88 |    4.12 |  3.38 |  3.88 |   1.75 |
> | Binary      |     1.9 |    4.2  |  3.8  |  3.1  |   2.0  |
> | Multiclass  |     3.86 |    3.71 |  2.71 |  2.57 |   2.14 |
> | All         |     2.44  |    4.04 |  3.36 |  3.2 |   1.96 |
>
> **Table R3**. Comparison against tree-based methods under tuned hyper-parameters
>
> For grid search in DL methods, we use fold 1 of each task for tuning the hyper-parameters. For each task, the training and test sets of each fold are fixed by OpenML task specification. The validation set is chosen as 10% of the training set, but fixed across different methods to ensure fairness. We use these fixed validation sets to select the tuned hyper-parameters.
>
> We will include the search grid in the appendix in the revised manuscript.
>
> **2. Support the Need of LM Path**
>
> There appears to be some confusion over the LM path of input embeddings. We would like to clarify this.
>
> Using the reviewer’s terminology:
>
>   **a.** The method named “FT” in Table 3 and Figure 6 refers precisely to what the reviewer termed as “2) existing embeddings” in the review.
>
>   **b.** In contrast, DisTab combines what the reviewer referred to as “1) LM embeddings” + “2) existing embeddings”.
>
> We agree with the reviewer that comparing 2) vs 1)+2) is the most relevant ablation to quantify the benefits of language augmentation. This is precisely why we reported it in the main paper. As already observed, the ablation demonstrates that language augmentation is beneficial overall. Please see our reply to Q3 in the initial response for performance discussion.
>
> **3. Limitations of GBDT models.**
> The reviewer asked several questions in their response and we will address them below.
>
> *a\. Why [9] was cited.*
>
> We used [9] to point out that header information can provide non-trivial additional information in tabular learning settings and that previous works have explored it. In particular, [9] states
>
> > the semantic meaning of one element can vary depending on the context composition. This formulation benefits the knowledge transfer across tables a lot.
>
> In our work, we used header information to directly improve task test performance.
>
> *b.\ Clarifying limitation of GBDT*
>
> Given the potential benefits offered by leveraging the header information, we are simply highlighting that, albeit appealing, it is neither clear nor trivial how to include unstructured input such as header information into GBDT models.
>
> *c\. Why [9] is not compared.*
>
> We note that TransTab [9] is not being compared for two reasons: a) It does not support the regression setting, so we could only report partial results. b) Xtab showed that TransTab underperforms state-of-the-art tabular DL methods. Since we adopted the same experimental settings and (a subset of) tasks from Xtab [1], reporting on TransTab is not informative.
>
> We also added an additional comparison against TP-BERTa, which also uses text headers to improve tabular learning. Please see our 2nd reply (Q5. Further comparison) to Reviewer 5FCu for further details.

---

> > ### Author Response · Authors · 2024-10-05
> > **Further Clarifications 2 of 2**
> >
> > **4\. The benefit of “pre-trained model can be directly used for inference”**
> >
> > There appears to be some confusion over the described benefits and we address it below.
> >
> > We do not claim that pre-trained models should be directly applied as the “production model” for a task. In fact, our results clearly show that fine-tuned models are better than pre-trained ones.
> >
> > The benefits we claimed are “meta-benefits” that streamline the practitioner’s job when performing model training and selection. On the one hand, since our pre-training models can be directly evaluated on the desired target objectives, model design and hyperparameter selection can be independent from fine-tuning. In contrast, when pre-training uses surrogate objectives, one cannot assess the quality of the resulting pre-trained model by itself and therefore pre-training becomes coupled with the downstream fine-tuning. This inevitably ends up complicating model design and selection.
> >
> > We thank the reviewer for pointing out this lack of clarity. We will add the above observation to the paper.
> >
> > **5\. Clarifying LM embedding.**
> >
> > The LM path is represented by $f_{enc}$ (cfr. (1)) in our text.  The path is not fixed, since it contains learnable projection layers (see Appendix B2 - Page 18). We thank the reviewer for pointing this out. We will move the definition of $f_{enc}$ to the main text and revise Figures 1 and 2 to highlight this learnable component.

---

> > > ### Comment · Reviewer_8cMY · 2024-10-14
> > > **Post-revision comments**
> > >
> > > The reviewer checked the updated revision and follow-up comments. Here are additional comments.
> > >
> > > ---
> > >
> > > ## Re: 1. Hyper-parameters choices
> > > The follow-up comment does not address the reviewer's concerns, specifically the reviewer's following points
> > > > - The reviewer would not consider evaluation setting used in two previous papers as a standard or a strong justification either. In this case, if the authors believe the setting is reasonable, they should explain why it's reasonable in their words.
> > > > - The authors argue that their reported results are "essentially equivalent" to those in [1], referring to the differences between their numbers and the numbers reported in [1]. But, this paper uses win matrices and average ranks as evaluation metrics, which are relative comparisons could significantly change even with $\pm 0.01$.
> > >
> > > The authors said
> > > > We also used a subset of the tasks used in [1]. Thus we politely disagree with the comment that “the experimental settings are different”.
> > >
> > > Using a subset of the tasks used in a previous work is not a convincing statement to justify the initial hyperparameters. Though the reviewer is not aware whether the hyperparameters were thoroughly tuned in the previous work, even if those were, those were tuned on the larger set of tasks. It may or may not be the best values for the subset used in this work.
> > >
> > > Apart from that, Appendix B.3 shows
> > > > We performed grid search for all methods to determine the key hyper-parameter values that have been
> > > reported in the previous section.
> > >
> > > However, the main result tables in the revision have not been updated while the authors did not tune hyperparameters before. It should be hard to believe that post-review hyperparameter tunings did not change any single results reported in the paper.
> > >
> > > ---
> > >
> > > ## Re: 2. Support the Need of LM Path + 5. Clarifying LM embedding.
> > >
> > > The reviewer thanks the authors for their further clarifications, but the updated revision does not address this point.
> > > Two suggestions:
> > > - "Language-Guided Embeddings" in Section 3.1 highlights differences from Wang & Sun 2022, citing their work
> > > -  Clarify exactly what is " language-guided embeddings" as in "The remaining models additionally include language-guided embeddings to provide additional context to input data.". Figure 1 says "Language-guided categorical (Top) and numerical (Bottom) embeddings for DisTab", which sounds like lookup-embedding is part of the language-guided embeddings, and it makes the added description (especially the word "additionally") implies there are two look-up embeddings in the model
> > >
> > > ---
> > >
> > > ## Re: 3. Limitations of GBDT models.
> > >
> > > The reviewer thanks the authors for the clarifications and wands them to update the revision with the statement (especially their answer for 3.c)
> > >
> > > ---
> > >
> > > ## Re: 4. The benefit of “pre-trained model can be directly used for inference”
> > >
> > > > This makes it easier to select which pre-trained model to employ for the downstream training. In
> > > contrast, previous pre-training strategies use surrogate objectives that are typically structurally different
> > > from the downstream task.
> > >
> > > The reviewer did not find any evidence to support this claim (as part of the claimed benefit). It seems more like just an opinion from the authors. If the reviewer's understanding is correct, when we want to choose some models among pre-trained models for downstream tasks (say BERT, ELECTRA, and DeBERTa), we can evaluate the pre-trained models for the downstream task and decide which model to further train. The pre-trained models' performance for the target tasks might be a proxy for model selection, so the authors could say "This might make it easier ..." but there is lack of evidence to say "This makes it easier ..."
> > >
> > > ---
> > >
> > > ## Minor
> > > - Table 1 did not fit to the TMLR format. It should be split into two tables or reformatted
> > > - Many of texts in the new Figs. 1 and 2 are very small and should be bigger as there are many white spaces.

---

> > > > ### Author Response · Authors · 2024-10-15
> > > > **Reply**
> > > >
> > > > We thank the reviewer for the feedback.
> > > >
> > > >
> > > > **Hyper-parameter choices**
> > > >
> > > >
> > > > In our previous replies, we explained that DL methods did undergo hyper-parameter tuning in the original manuscript. We simply added the information on the search grid to the revised manuscript and the results are thus unchanged.
> > > >
> > > >
> > > > For tree-based methods, we performed the grid search as the reviewer requested. We ended up with XgBoost, LightGBM and CatBoost having the same hyper-parameters as those from the original manuscript, thus no change to the results.
> > > >
> > > >
> > > > Random forests obtained a different set of hyper-parameters and were updated in the revised manuscript. We apologize for missing the update to the other tables and have included the new results.
> > > >
> > > >
> > > > **Limitation of GBDT model**
> > > >
> > > >
> > > > The limitation of GBDT was already revised in the manuscript (Page 3 last paragraph).
> > > >
> > > >
> > > > **Clarifying LM Path**
> > > >
> > > >
> > > > The key difference from Wang & Sun 2022 is that we use a single token for cell values/column headers, instead of a series of tokens in Wang & Sun 2022. For instance, “Job: doctor” is mapped to a single token (via average pooling) instead of multiple tokens (“Job”, “:” and “doctor”). The motivation is explained by the observation that tables should be permutation invariant with respect to its columns.
> > > >
> > > >
> > > > How LM guidance is combined with existing embeddings is formally defined in (1) and (2), as well as illustrated in Fig 1. The revised explanation is already present in the manuscript (Page 5 first paragraph).
> > > >
> > > >
> > > > Specifically to the reviewer’s question, there is no 2nd look-up embedding. The column header and cell-value are processed by a LM, pooled and lastly projected with an learnable MLP to derive the language-guided embedding.
> > > >
> > > >
> > > > **Benefits of distillation pre-training**
> > > >
> > > >
> > > > We thank the reviewer and have changed the wording as suggested.

---

### Review · Reviewer_5FCu · 2024-10-01

**Summary Of Contributions:**

The paper introduces DisTab, a deep learning framework for tabular data that incorporates model distillation from tree-based models and language guidance for feature embeddings. Distillation leverages the strengths of tree-based models during pretraining, while language guidance adds semantic context from textual data, such as column headers.

**Audience:**

Yes

**Claims And Evidence:**

Yes

**Requested Changes:**

Please see the weaknesses part.

[A] Jiahuan Yan, Bo Zheng, Hongxia Xu, Yiheng Zhu, Danny Z. Chen, Jimeng Sun, Jian Wu, Jintai Chen: Making Pre-trained Language Models Great on Tabular Prediction. ICLR 2024

[B] Chao Ye, Guoshan Lu, Haobo Wang, Liyao Li, Sai Wu, Gang Chen, Junbo Zhao: Towards Cross-Table Masked Pretraining for Web Data Mining. WWW 2024: 4449-4459

[C] Zifeng Wang, Jimeng Sun: TransTab: Learning Transferable Tabular Transformers Across Tables. NeurIPS 2022

[D] Bingzhao Zhu, Xingjian Shi, Nick Erickson, Mu Li, George Karypis, Mahsa Shoaran: XTab: Cross-table Pretraining for Tabular Transformers. ICML 2023: 43181-43204

[E] Geoffrey E. Hinton, Oriol Vinyals, Jeffrey Dean: Distilling the Knowledge in a Neural Network. CoRR abs/1503.02531

**Strengths And Weaknesses:**

#### Strengths:

- **Design of the Methodology**: The integration of distillation from tree-based models to pretrain a DL model is a new approach for a set of heterogeneous tabular datasets.
- **Performance**: DisTab showcases strong performance against a range of existing DL models and even matches or surpasses GBDTs, which have historically been more successful in tabular data tasks.
- **Versatility**: The model is capable of handling various tabular data tasks, including regression, binary classification, and multi-class classification.

#### Weaknesses:

- **Unclear Explanation of Distillation**: The paper states on the third page, fifth line, that "Our proposed DisTab also adopts domain-specific pre-training but selects knowledge distillation as the primary pre-training objective." However, it remains unclear whether this knowledge distillation involves only using CatBoost to label real and synthetic data, or if it includes a more traditional knowledge distillation [E] process during pretraining.
- **Dependence on Textual Data**: The language-guided embeddings rely on the availability of textual data, such as column headers or descriptions. In datasets where such information is sparse or absent, this advantage may be diminished.
- **Lack of Comprehensive Comparisons**: The paper does not compare DisTab with other recent methods that integrate language models, such as **TP-BERTa** [A], **CM2** [B], and **TransTab** [C], which also aim to enhance tabular learning through the use of textual data.
- **Lack of Comparison with Other Data Augmentation Techniques**: DisTab uses synthetic data to augment the training set during pretraining. However, the paper does not compare its approach with other data augmentation methods for tabular data, which could also be applied in the same pre-training context.
- **More Rigorous Experimental Design**: The experimental design could be improved by separating the pretraining and fine-tuning datasets, similar to how **Xtab** [D], **TP-BERTa** [A], and **CM2** [B] structure their experiments. By using different datasets for pretraining and finetuning, DisTab's ability to generalize across different domains could be more thoroughly tested. This approach would better showcase the potential of DisTab, especially in scenarios involving transfer learning, and would align it with the experimental standards of comparable models in the field.

---

> ### Author Response · Authors · 2024-10-05
> **Reply 1 of 2**
>
> We thank the reviewer for the feedback. Please find our response below.
>
> **1\. Clarifying distillation**
>
> We use CatBoost solely to label real and synthetic data during pre-training. This is formally defined in Section 3.2 Eq. 5, where the teacher model $g_T$ (defined at the end of Page 5) labels both real and synthetic samples. The fully teacher-labeled dataset is used for pre-training. We will further clarify this difference with traditional distillation methods in the revised manuscript.
>
> **2\. Dependence on Textual Data**
>
> We agree with the reviewer that textual headers are necessary for language guidance. We wish to highlight that our method provides a performance advantage *when* headers are present and that tabular headers are *often* available in practice.
>
> Moreover, we point out that tabular data is a human construct and headers are often necessary even for human interpretation (e.g. human-readable spread-sheets). It is thus reasonable to assume the presence of meaningful headers and to allow ML models access to them. Absent headers due to data privacy is outside the scope of the current work.
>
> We will add the above points to contextualize our proposed technique in the revised manuscript.
>
> **3\. Other data augmentation techniques**
>
> We respectfully disagree with the comment that other augmentation techniques are not compared. Augmentation techniques are usually coupled with the target pre-training objectives, such as data masking for reconstruction pre-training, or data perturbation for contrastive pre-training. These augmentation techniques are present in the pre-training procedures of baseline methods (e.g., Saint, Scarf, Vime) and indeed used in the experiments. Section 4.1 thus reports comparison with different pre-training objectives along with their augmentation techniques. We will leave ablations integrating alternative data augmentation techniques within DisTab to future works, since the current work focuses on the benefits of language guidance.
>
> **4\. Experimental Design**
>
> We thank the reviewer for the suggestion. Investigating cross-table transfer and pre-training is one of our future directions. It is orthogonal to our current work and could be directly combined.
>
> However, the current work is not aimed at demonstrating cross-tabular transfer. Rather, we show that pre-training can be performed without surrogate datasets/tasks. We leverage tree-based distillation to address overfitting on small tabular datasets and to mimic the favorable inductive bias from tree-based methods. Our experiments demonstrate that this is a viable approach and is in fact more robust than cross-tabular pre-training, as performed in XTab.

---

> > ### Author Response · Authors · 2024-10-05
> > **Reply 2 of 2**
> >
> > **5\. Further comparison**
> >
> > We thank the reviewer for the additional references and will include them in the revised manuscript. We note that TransTab is not being compared for two reasons: **a)** It does not support the regression setting, so we could only report partial results. **b)** XTab showed that TransTab underperforms state-of-the-art tabular DL methods. Since we adopted the same experimental settings and (a subset of) tasks from Xtab, reporting on TransTab is not informative.
> >
> > As per the reviewer’s suggestion, we evaluated the performance of TP-Berta in our setting. Due to limited time and computational resources we report results over a subset of tasks. We observe that similar to XTab, cross-tabular transfer as pre-training does not guarantee robust test performance.
> >
> > | Task                             | Metric   |   TP-Berta |       Ours |
> > |---------------------------------|---------|-----------|-----------|
> > | abalone                          | rmse     |     2.2882 |     2.1725 |
> > | boston                           | rmse     |     4.2931 |     2.4388 |
> > | house_prices_nominal             | rmse     | 26088.8    | 23204.2    |
> > | blood-transfusion-service-center | roc_auc  |     0.5843 |     0.7416 |
> > | churn                            | roc_auc  |     0.8773 |     0.9194 |
> > | credit-g                         | roc_auc  |     0.7534 |     0.7729 |
> > | kc1                              | roc_auc  |     0.7868 |     0.8066 |
> > | pc4                              | roc_auc  |     0.8645 |     0.9503 |
> > | qsar-biodeg                      | roc_auc  |     0.6641 |     0.9224 |
> > | car                              | acc      |     0.9919 |     0.9954 |
> > | segment                          | acc      |     0.9229 |     0.9446 |
> > | wine-quality-white               | acc      |     0.6543 |     0.6612 |
> >
> > Finally, as noted in our reply to Q4 above, cross-tabular training is one of our future directions and it could be directly integrated with our approach. We will further discuss this in our conclusion.

---

> > > ### Comment · Reviewer_5FCu · 2024-10-10
> > >
> > > Thank you for your response, clarifications, and for providing the additional experiments. I appreciate the effort in addressing the concerns raised.
> > >
> > > However, I would like to request further insights regarding the role of the language model in DisTab. Specifically:
> > >
> > > **Detailed Performance Analysis:** In the ablation study, the results in the last two rows show minimal differences in average rank.  Could you provide more detailed results and the following information:
> > >
> > > - Which datasets benefit more from the use of the language model?
> > > - What are the specific characteristics of these datasets (i.e., those where the language model improves performance)?
> > >
> > > **Language-Guided Embedding:** Could you elaborate on the properties of the learned Language-Guided Embedding? Understanding its characteristics would help clarify the contribution of the language model.
> > >
> > > **Validation with Meaningless Feature Names:** To further validate the role of the language model, could you test DisTab on datasets with neutral or meaningless feature names, such as 'Feature One', 'Feature Two', or 'A', 'B', etc.? This would help confirm whether the language model is truly providing meaningful guidance during the learning process.
> > >
> > > Thank you once again for your work, and I look forward to your response.

---

> > > > ### Comment · Action_Editor_sMx3 · 2024-10-14
> > > >
> > > > Dear Reviewer 5FCu
> > > >
> > > > Thank you for your responses to the authors' reply.
> > > >
> > > > Could you let me know whether the authors' reply ([No.1](https://openreview.net/forum?id=p6KIteShzf&noteId=m9Pu0oTYPU), [No.2](https://openreview.net/forum?id=p6KIteShzf&noteId=uPkW8336Kg)) solve your questions in your [initial review](https://openreview.net/forum?id=p6KIteShzf&noteId=AJY10V9APd), in particular, about **Lack of Comprehensive Comparisons** and **Lack of Comparison with Other Data Augmentation Techniques**?
> > > >
> > > > Best,
> > > > AE

---

> ### Author Response · Authors · 2024-10-15
> **Reply**
>
> We thank the reviewer for the feedback. Please see our replies below.
>
> **Detailed performance analysis**
>
> We agree with the reviewer that average rank misses out more fine-grained comparison. They are primarily used to give the overall standing of a method, and to be consistent with previous works that used the same metric. This is why we also reported win matrices in Fig 5 for pairwise comparison. The results show that DisTab outperforms FT in 17 out of 25 tasks.
>
> **What datasets performed better**
>
> We observe that language guidance tends to improve tasks with higher proportions of categorical features with strong textual meanings (e.g., okcupid-stem dataset that contains online dating profiles). This observation is consistent with what was observed in the TP-BERTa paper. We note that this is a post-hoc qualitative analysis and we leave the systematic investigation to future works.
>
> **Characterizing the language embeddings**
>
> We hypothesize that language embeddings could influence the attention mechanism and provide different feature mixing pathways for the trained model. To validate this, we visualize the attention map for DisTab vs. FT. The maps show that while the two model variants identify similar key features, language guidance provides DisTab with slightly different feature mixing in the trained model. Please see Appendix C.2 for the attention activation visualization.
>
> We note that systematically characterizing the mechanism of language embeddings is beyond the scope of the current work, as we focus on quantitative performance of our approach on realistic datasets. Systematic characterization of language embeddings will likely require more fine-grained control over the training data, including the use of synthetic datasets (such as in [1, 2]) where the ground-truth model for a table is known and the amount of noise (labeling, missing values, confounding columns) could be directly controlled.  We leave this as a future work and thank the reviewer for the suggestion.
>
> [1] Why do tree-based models still outperform deep learning on typical tabular data, Grinsztajn et al.
>
> [2] Revisiting Deep Learning Models for Tabular Data, Gorishniy et al.
>
> **Dummy textual data**
>
> Following the reviewer's suggestion, we replaced the textual input with random strings in okcupid-stem, steel-plates-fault, abalone and moneyball datasets and retrained them with dummy textual data. We performed 5 runs for this setup - sampling a different set of random textual data for each trial - to reduce the potential impact of random fluctuations and obtain a more robust estimate of the model's behavior."
>
>
> |  Dataset           | Metric | Ours | Dummy Text |
> |--------------------|---|---|------------------|
> | okcupid-stem  |     Acc | 76.2 | 75.4             |
> | steel-plates-fault | Acc | 81.0 | 80.3             |
> | abalone            | RMSE | 2.17 | 2.20             |
> | moneyball         | RMSE | 21.8 | 22.1             |
>
>
> The results suggest that proper textual input provides meaningful guidance during training.

---

### Review · Reviewer_JBy8 · 2024-10-01

**Summary Of Contributions:**

This paper focuses on learning tabular data. It proposes to distill a tree-based model into a tabular transformer, and to incorporate auxiliary textual information from the column headers into the token embedding. The method outperforms existing DL and tree-based methods on various tabular datasets.

**Audience:**

Yes

**Claims And Evidence:**

Yes

**Requested Changes:**

1.	Could you further explain how the proposed architecture achieves permutation invariance? Is it implemented simply by removing the positional encoding?

2.	The value of $k$ in Equation (3) should be specified.

3.	Could you explain what is the dimension of the numerical embedding? It seems that Equation (2) concatenates several $d$-dimensional vector together, so that the embedding is a $|\Sigma| d$-dimensional vector?

4.	Minor. The last line on Page 3 is not complete.

**Strengths And Weaknesses:**

Strengths:

1.	The paper is well written and easy to follow.

2.	The experiments are conducted on comprehensive tabular datasets.

3.	The paper conducts sufficient ablation studies.


Weaknesses:

1.	It is not very convincing that incorporating textual information from the column headers consistently leads to a better performance. If we look at the rows Base and Base+LM in Table 3, we find that incorporating textual information actually degrades the performance. However, if distillation from tree-based model is used, the textual information improves the performance. This nuance suggests that there must be something in the distillation process that makes the textual information more useful than they should be, and the authors are expected to further explore this mechanism (or at least provide initial discussions and experiments).

2.	The performance gain of the proposed method is mainly attributed to the distillation from a SOTA tree-based model, according to the ablation in Table 3. However, using distillation from a good model (good performance, good representation) to improve performance is not a novel idea. I’m concerned that this method does not provide new insights. It would be better if the paper further analyzes for which type of tasks/data the distillation significantly boosts the performance of the student model, and for which type of tasks/data the distillation even makes it worse.

3.	The metric Average Rank, though being used by previous works, only focuses on ranks and ignores the exact margin of performance gain between different methods.

---

> ### Author Response · Authors · 2024-10-05
> **Reply 1 of 2**
>
> **1\. Drop in performance for Base + LM vs. Base**
>
> We agree with the reviewer that this is an interesting behavior. This happens mainly because DL architectures are prone to overfitting, especially on smaller datasets (this is true in general for DL models, and we discussed this in Section 3.2 for tabular data). Pre-training is precisely designed to mitigate this problem. Without pre-training, the addition of the LM embeddings introduces additional degrees of freedom to a model already struggling with insufficient data, hence worsening overfitting. In contrast, pre-training provides an abundance of synthetic data to prevent overfitting, leading to more robust performance. We will add this observation to the discussion of Table 3.
>
> Specifically, Base vs LM+Base perform nearly identically in the majority of tasks, where performance drop, if any, is minimal. LM+Base lost out in tasks of smaller sizes, thus leading to worse rankings. For a summary, we report in Table R1 below the average task performance of Base, Base+LM and DisTab. (Please see reply to Q3 below for details on average performance computation.)
>
> |  | Base | LM+Base | DisTab |
> | --- | --- | --- | --- |
> | Regression | 0.781 | 0.781 | 0.809 |
> | Binary | 0.854 | 0.851 | 0.858 |
> | Multiclass | 0.803 | 0.801 | 0.821 |
>
> **Table R1**. Average performance comparison between Base, Base+LM and DisTab
>
> The results show that with respect to average performance, Base vs LM+Base are comparable, with Language guidance not degrading the performance. In contrast, DisTab noticeably outperforms the baselines by incorporating both pre-training and language guidance.
>
> **2\. New insights**
>
> We agree with the reviewer that model distillation as a general concept is not new. However, we highlight that **1)** no existing work on tabular DL proposes distillation from tree-based methods. Rather, most of the current literature on tabular DL (e.g., Saint, Xtab, VIME) focuses on other pre-training approaches. **2)** Addressing the performance gap between tabular DL and tree-based methods is considered an open and relevant question [1, 2]. We also commented on why existing pre-training approaches may not be as robust (see 2nd paragraph of Section 3.2). Therefore, we argue that showing that distillation is a key contributor to achieving state-of-the-art performance with tabular DL is novel and relevant.
>
> Additionally, we argue that our results are more nuanced than “mainly attributed to the distillation from a SOTA tree-based model”. In fact, we explicitly stated in the first paragraph of Page 9
>
> > The results suggest that distillation of tree-based methods alone may be insufficient to reliably improve tabular DL models. However, FT, which combines distillation and fine-tuning, clearly outperforms Base, winning 20 out of the 25 tasks.
>
> Our win-matrices in Fig 6 implicitly present some results requested by the reviewer. It shows that distillation alone does not outperform the teacher model in all three settings (less than half of tasks improved). This may be expected as pre-training only uses synthetic data and labels, which differs from the actual training distribution. Fine-tuning on the real data is thus key to correct this distribution mismatch and achieves much more robust results. Fig 6 shows that across all settings, distillation along with fine-tuning consistently improves model performance consistently in 20/25 tasks.
>
> [1] Why do tree-based models still outperform deep learning on typical tabular data, Grinsztajn et al.
>
> [2] Trompt: Towards a better deep neural network for tabular data, Chen et al.
>
> **3\. Average rank metric**
>
> We thank the reviewer for the suggestion. To compute average performance across tasks, we transform RMSE for regression tasks to R2 score, such that its range is [0, 1] and larger the better, aligning with AUC and accuracy used for binary and multi-class classification tasks. This allows us to average task performance across different task settings.
>
> We report the average task performance in Table R2 below, using the above metric. The table shows similar results as those reported in the paper, DisTab is able to match the performance of CatBoost overall and outperform the others. In the multi-class setting, DisTab noticeably outperforms CatBoost and matches the results of GBM and XGB.
>
>
> |   | CAT | RF | GBM | XGB | DisTab |
> | --- | --- | --- | --- | --- | --- |
> | Regression | 0.81 | 0.784 | 0.795 | 0.795 | 0.809 |
> | Binary | 0.86 | 0.849 | 0.85 | 0.854 | 0.858 |
> | Multi-class | 0.813 | 0.812 | 0.821 | 0.822 | 0.821 |
> | All | 0.831 | 0.818 | 0.824 | 0.826 | 0.832 |
>
> **Table R2**. Average task performance comparison between DisTab and tree-based methods.
>
> Since aggregating metrics across tasks might miss nuances in the behavior between different methods, the individual absolute task performance is reported in the Appendix Table 16 (Page 19) for a full “lossless” comparison.

---

> > ### Author Response · Authors · 2024-10-05
> > **Reply 2 of 2**
> >
> > **4\. Permutation invariance.**
> >
> > Yes. Permutation invariance is achieved by not using positional encoding.
> >
> > **5\. Value of K and dimension of numerical embeddings**
> >
> > We concatenate three random features of different bandwidth together to form numerical embeddings. Thus k = d/3 where d is the dimension of an input token. We will clarify this in the revised manuscript.

---

> > ### Comment · Reviewer_JBy8 · 2024-10-24
> >
> > I would like to thank all authors for the response. Most of my concerns are addressed accordingly, and I will take into account new discussions/experiments added to the revised manuscript when making my recommendation.

---

### Author Response · Authors · 2024-10-10
**Summary of Changes & Revised Manuscript**

We thank the reviewers for their feedback and discussion. We have revised the manuscript to reflect the requested changes and any clarification to the reviewers’ questions. The revision is highlighted in blue in the new manuscript. We summarize the key revisions below.

[Reviewer 5FCu] We:
1. Clarified how distillation is performed in pre-training.
2. Discussed the usage of textual data for tabular data.
3. Performed additional experiments on TP-BERTa and included the results.

[Reviewer JBy8] We:
1. Discussed the performance comparison between Base vs. Base+LM, to address the usage of language guidance without pre-training.
Highlighted how distillation contributes towards tabular DL and discussed its relations with existing techniques.
2. Added average performance comparison to reflect the scale of performance comparison between DisTab and tree-based methods.
3. Clarified the usage of random features in numerical embeddings.

[Reviewer 8cMY] We:
1. Added hyper-parameter tuning details in the appendix.
2. Clarified the input embeddings used in the model variants used for the ablation study. This aims to further emphasize the comparison between “existing embeddings” vs. “existing embeddings + language” embeddings, to demonstrate the effectiveness of language guidance.
3. Clarified the “flexibility” and in general the potential advantages of DL with respect to tree-based methods.
4. Further discussion on the motivation of our approach.
5. Included an end-to-end model architecture diagram.
6. Further clarified the language embedding structure in the main text.
7. Clarified the advantage of using distillation pre-training.

We are happy to address any further comments/questions the reviewers may have.

---

### Author Response · Authors · 2024-11-21
**Thank you**

Dear AE and Reviewers,

We wish to thank the AE and reviewers for their insights, suggestions and engagements during the review process, which have helped us to improve our work.

We have submitted the camera ready version with all requested changes.

Best regards,
Authors

---

> ### Comment · Action_Editor_sMx3 · 2024-11-24
> **Confirmation of changes in camera-ready version**
>
> I appreciate the authors for submitting the camera ready version. I found several changes that could affect the contents of the manuscript. I want to confirm whether the following changes are intentional:
> - p5: $p_\sigma(x) p_\sigma(y) \approx \exp(x-y)/σ^2$ in the draft has been changed to $p_\sigma(x) p_\sigma(y) \approx \exp(\sigma^2(x-y)^2/2)$
> - p5: the coefficient of $2\pi$ has been removed from the definition of $v$ in Eq. (2)
> - p5: The reference to Li et al. (2021) and Gorishniy et al. (2022) for Eq. (2) have been removed.
>
> Best,
> AE

---

> > ### Author Response · Authors · 2024-11-24
> > **Confirmation of changes**
> >
> > Dear AE,
> >
> > Thank you for the question. We confirm that the changes are intentional.
> >
> > For the numerical embedding, $2\pi$ can be absorbed as part of the $\sigma^2$, and approximates Gaussian kernel $p_\sigma(x) p_\sigma(y) \approx \exp(\sigma^2(x-y)^2/2)$. We apologize for the typo in the definition of the Gaussian kernels in the previous iterations.
> >
> > Gorishniy et al. (2022) is still referenced after Eq. 3 in relation to numerical embedding. Li et al. (2021) is not referenced as the original use of such numerical embedding, to our knowledge, was Rahimi & Recht (2007), which is referenced fo  Eq. 2 in the current revision.
> >
> > We are happy to further clarify if needed.
> >
> > Best,
> > Authors

---

> > > ### Comment · Action_Editor_sMx3 · 2024-11-24
> > > **Thank you**
> > >
> > > Thank you for the responses. They clarified my questions.

---

### Decision · Action_Editor_sMx3 · 2024-11-07

**Recommendation:** Accept with minor revision

**Comment:**

I suggest the authors follow the reviewers’ suggestions and revise the draft. In particular, the following points:

- While this method of evaluation by selecting the hyperparameter is reasonable, reviewers commented that more detailed explanations of the reasons for choosing this protocol are preferable.
- It was pointed out that the word *fold* in the previous version may cause confusion. It is suggested that a different wording be used in Section B.3.
- Σ in Eq. (2) is not defined. Its explanation should be given.
- The last line of the **Notation** paragraph in Section 3 is incomplete and should be corrected appropriately.

**Audience:**

Applying DL methods and language models to tabular data is one of the popular topics in tabular learning research. In particular, several studies have compared GBDT and transformer-based models and used side information such as headers using language models. This paper follows these lines of research and is of interest to some audiences in the TMLR community.

**Claims And Evidence:**

This paper studies tabular data learning using transformer-based models. It proposes a model that constructs embeddings using column information with language models, distills knowledge from tree-based models, and augments the datasets by mixup. The authors claim to achieve predictive performance comparable to the GBDT models. Ablation studies are also conducted to elucidate the contribution of each model component.

Reviewers expressed valid concerns about the evaluation methods in the numerical experiments, particularly the hyperparameter tuning procedures. Reviewers also noted that the proposed method is an incremental improvement and could benefit from more comprehensive ablation studies, especially about mixups and language model selection.

Indeed, choosing a single set of hyperparameters for multiple benchmark datasets based on their average performances, as the paper does, is not the most standard common practice. However, the evaluation protocol, including the hyperparameter choice, is reasonable for machine learning tasks. Therefore, despite these weak points, the experiments generally support the claims as long as the authors explicitly describe the above limitations derived from the in-depth experiments.